# Learning Rate Scheduling with Matrix Factorization for Private Training

## Abstract

We study differentially private model training with stochastic gradient descent under learning rate scheduling and correlated noise. Although correlated noise, in particular via matrix factorizations, has been shown to improve accuracy, prior theoretical work focused primarily on the prefix-sum workload. That workload assumes a constant learning rate, whereas in practice learning rate schedules are widely used to accelerate training and improve convergence. We close this gap by deriving general upper and lower bounds for a broad class of learning rate schedules in both single- and multi-epoch settings. Building on these results, we propose a learning-rate-aware factorization that achieves improvements over prefix-sum factorizations under both MaxSE and MeanSE error metrics. Our theoretical analysis yields memory-efficient constructions suitable for practical deployment, and experiments on CIFAR-10 confirm that schedule-aware factorizations improve accuracy in private training.

## 1 Introduction

Privacy has become a major concern as machine learning systems are trained on sensitive data such as personal communications, financial transactions, and medical records. Beyond the risk of direct data exposure, models themselves may memorize and unintentionally reveal private information, creating serious ethical and security challenges. These concerns are especially pressing for production-level large language models trained on vast and heterogeneous datasets.

A widely studied approach to mitigating these risks is differential privacy (DP), which provides formal mathematical guarantees that the output of a learning algorithm does not reveal sensitive information about any individual training example (Dwork et al., 2006). In practice, DP is often achieved by injecting carefully calibrated noise into either the gradients, ensuring that an adversary cannot infer the presence or absence of a single data point with high confidence. More recently, large-scale efforts such as VaultGemma (VaultGemma Team, 2025) have demonstrated that it is possible to train billion-parameter models with rigorous privacy guarantees, showing that DP can be integrated into state-of-the-art architectures without prohibitive utility loss.

To make model training differentially private, algorithms typically inject noise into the gradients to mask the contribution of any individual data point. The most common approach, DP-SGD, adds independent Gaussian noise at each update, which provides strong privacy guarantees but can significantly reduce accuracy (Abadi et al., 2016). *Matrix factorization* has emerged as a more general alternative that introduces correlations in the injected noise, enabling improved accuracy while preserving privacy (Choquette-Choo et al., 2023a;b).The approach has also seen practical adoption, with Google reporting its use for training production on-device language models in their 2024 blog post "Advances in private training for production on-device language models" (Xu & Zhang, 2024).

Recent work has focused on making matrix factorization memory efficient (McKenna, 2025; Andersson & Pagh, 2025; Kalinin et al., 2025; McMahan et al., 2024), and it has also been analyzed theoretically, mostly in the setting of Toeplitz workloads (Fichtenberger et al., 2023; Henzinger et al., 2024; Henzinger & Upadhyay, 2025; Dvijotham et al., 2024). However, existing utility analyses assume a constant learning rate. While Denisov et al. (2022) introduced a non-Toeplitz workload with varying learning rates, its theoretical properties remain largely unexplored. In this work, we address this gap by studying matrix factorization under learning rate schedules.

Learning rate scheduling plays a critical role in the optimization of machine learning models, and a variety of strategies have been proposed in the literature. Popular approaches include cosine annealing (Loshchilov & Hutter, 2017) and cyclical learning rates (Smith, 2017), which adapt the step size during training to improve convergence. Another common technique is warm-starting (He et al., 2016), where models begin with a small learning rate that is gradually increased, as used in large-scale training setups (Goyal et al., 2017). In this work, we focus on learning rate decays available in PyTorch Paszke et al. (2019) such as exponential, linear, polynomial and cosine, which are widely used and can be easily applied in practice.

Learning rate scheduling can be particularly useful in private training when the number of iterations is limited. By accelerating convergence, it enables higher accuracy in settings such as warm-up training (Kurakin et al., 2022), private fine-tuning (Luo et al., 2021), and training under computational constraints. It has also been combined with matrix factorization as a form of learning rate cool-down (Choquette-Choo et al., 2023b;a; 2025), and was shown to provide improvements over fixed learning rates in Denisov et al. (2022), where the workload of our interest was originally introduced.

**Contribution**

- We theoretically analyze the problem of matrix factorization under learning rate scheduling. We establish general lower and upper bounds for MaxSE and MeanSE in single-epoch, as well as for MeanSE in multi-epoch for a large class of schedulers. Here, MaxSE characterizes the maximum variance of the added noise, while MeanSE captures the average variance across iterations.

- We propose a learning-rate-aware Toeplitz factorization, which for exponentially decaying learning rate is provably optimal in MaxSE under single-epoch and improves upon the proposed upper bound for MeanSE. We adopt this factorization for memory efficient, multi-epoch training by making it banded inverse.

- We show numerically that the proposed factorization is close to optimal in all metrics.

- We show experimentally on CIFAR-10 that banded inverse factorizations benefit from learning rate scheduling. Moreover, we demonstrate that the proposed learning-rate-aware factorization achieves even further accuracy improvements.

## 2 BACKGROUND

The most common way to train differentially private models is by using *DP-SGD* (Abadi et al., 2016). At each step we receive a gradient $g_i \in \mathbb{R}^d$, clip it to a fixed $\ell_2$ norm $\zeta > 0$, and add appropriately scaled independent Gaussian noise $z_i \sim \mathcal{N}(0, I_d \, \sigma^2)$, where $\sigma$ depends on the target privacy level $(\epsilon, \delta)$. The model is then updated as

$$\theta_i = \theta_{i-1} - \eta_i \big( \text{clip}(g_i, \zeta) + z_i \big), \tag{1}$$

where $\eta_i$ is the learning rate at step $i$.

This procedure can be improved by correlating the noise across iterations. To formalize this, we define a matrix $G \in \mathbb{R}^{n \times d}$ of stacked gradients, a matrix $\Theta \in \mathbb{R}^{n \times d}$ of intermediate models, and a workload matrix $A \in \mathbb{R}^{n \times n}$ that encodes the training process such that $\Theta = AG$. If we use a constant learning rate, this matrix, denoted $A_1$, is a lower triangular matrix of ones. For varying learning rates we instead use a matrix $A_\chi$, described later.

To ensure that the intermediate models are differentially private, we can apply a matrix factorization mechanism. Specifically, we factorize $A$ into two matrices $B$ and $C$, then compute $CG$, add Gaussian noise to ensure differential privacy, and finally multiply the result by $B$ as post-processing:

$$\widehat{AG} = B(CG + Z) = A(G + C^{-1}Z), \tag{2}$$

which is equivalent to adding correlated Gaussian noise with covariance structure induced by $C^{-1}$ to the gradients.

The remaining question is: *how much noise must be added, and does this procedure remain differentially private when the gradients are not known in advance but depend on the current model?* The foundational work of Denisov et al. (2022) shows that the procedure is indeed differentially private, even when the gradients adaptively depend on the model, provided we add noise of scale

$$\sigma = \zeta \cdot \sigma_{\epsilon,\delta} \cdot \text{sens}(C) = \zeta \cdot \sigma_{\epsilon,\delta} \cdot \|C\|_{1 \to 2}, \tag{3}$$

---

**Algorithm 1** Differentially Private SGD with Matrix Factorization and Learning Rate Schedules

---

**Require:** Model initialization $\theta_0 \in \mathbb{R}^d$, dataset $D$, batchsize $B$, model loss $\ell(\theta, d)$, clipnorm $\zeta > 0$,
    learning rate $\eta$, correlation matrix $C \in \mathbb{R}^{n \times n}$, learning rate scheduler $\chi_i$,
    noise matrix $Z \in \mathbb{R}^{n \times d}$ with i.i.d. entries $\mathcal{N}(0, \text{sens}^2(C)\sigma_{\epsilon,\delta}^2 \zeta^2)$.
    **for** $i = 1, 2, \ldots, n$ **do**
        $S_i \leftarrow \{d_1, \ldots, d_B\} \subseteq D$   (select a data batch)
        $g_i \leftarrow \nabla_\theta \ell(\theta_{i-1}, d_j)$   for $j = 1, \ldots, B$
        $x_i \leftarrow \sum_{j=1}^{B} \min(1, \zeta/||g_j||) \cdot g_j$   (clip gradients)
        $\hat{x}_i \leftarrow \frac{1}{B}\left(x_i + [C^{-1}Z]_{[i,\cdot]}\right)$
        $\theta_i \leftarrow \theta_{i-1} - (\chi_i \eta)\hat{x}_i$
**Ensure:** $\theta_n$

---

where $\zeta$ denotes the clipping norm, and $\sigma_{\epsilon,\delta}$ is the noise multiplier of the standard Gaussian mechanism, which can be computed numerically (Balle & Wang, 2018). The term $\text{sens}(C)$ represents the global sensitivity of the Gaussian mechanism for the product $CG$ when the row or rows corresponding to a single datapoint in $G$ change; it can be computed explicitly as $\|C\|_{1\rightarrow2}$, the maximum column norm of $C$. The case of multi-participation (multi-epoch) is discussed in Section 4.1. Now we have all the steps to train the model with differential privacy as presented in Algorithm 1.

The choice of factorization $A = BC$ significantly impacts the quality of the private estimation. Following the work of Denisov et al. (2022); Henzinger et al. (2024) we quantify the *approximation quality* by either the mean squared error (MeanSE) or the maximum expected squared error (MaxSE), which can be computed as

$$\text{MeanSE}(B, C) = \sqrt{\frac{1}{n} \mathbb{E}_Z \|AG - \widehat{AG}\|_F^2} = \frac{1}{\sqrt{n}} \|B\|_F \|C\|_{1\rightarrow2} \, \sigma_{\epsilon,\delta} \, \zeta, \tag{4}$$

$$\text{MaxSE}(B, C) = \mathbb{E}_Z \|AG - \widehat{AG}\|_\infty = \|B\|_{2\rightarrow\infty} \|C\|_{1\rightarrow2} \, \sigma_{\epsilon,\delta} \, \zeta,$$

where $\|\cdot\|_F$ denotes the Frobenius norm and $\|\cdot\|_{2\rightarrow\infty}$ the maximum row $\ell_2$-norm. These approximation errors are independent of $G$, and the term $\sigma_{\epsilon,\delta}\zeta$ is independent of the matrix factorization. To isolate the contribution of the factorization $(B, C)$, we will use the notation $\text{MeanSE}(B, C)$, $\text{MaxSE}(B, C)$ assuming $\zeta = \sigma_{\epsilon,\delta} = 1$ in the theoretical analysis.

## 3 METHOD

We now turn to the workload of stochastic gradient descent (SGD) with learning rate scheduling. Let $\chi_1, \chi_2, \ldots, \chi_n$ be a sequence with $\min \chi_t = \beta > 0$ and $\max \chi_t = 1$, representing a learning rate scheduler such that the actual learning rate at time $t$ is $\eta_t = \eta \chi_t$. We assume that $\beta$ is reasonably separated from 1, as the regime $\beta \rightarrow 1$ is not of interest since it nullifies the benefits of scheduling. Then the workload matrix of interest is:

$$A_\chi = \begin{pmatrix} \chi_1 & 0 & 0 & \cdots & 0 \\ \chi_1 & \chi_2 & 0 & \cdots & 0 \\ \chi_1 & \chi_2 & \chi_3 & \cdots & 0 \\ \vdots & \vdots & \vdots & \ddots & \vdots \\ \chi_1 & \chi_2 & \chi_3 & \cdots & \chi_n \end{pmatrix} = A_1 \times D, \tag{5}$$

where $A_1$ is a prefix-sum matrix (lower triangular matrix of all ones) and $D$ is a diagonal matrix of learning rates, i.e., $D = \text{diag}(\chi_1, \ldots, \chi_n)$. We will study the problem of *optimal matrix factorization* in MaxSE and MeanSE metrics for the matrix $A_\chi$ with the **learning rate decays** given in Table 1. For the experiments, we will also include the constant learning rate ($\chi_k = 1$).

In this work, we prove general lower and upper bounds for the MaxSE and MeanSE errors. For the upper bound, we use a prefix-sum-based factorization given by $B = A_\chi(A_1)^{-1/2}$ and $C = A_1^{1/2}$, which has been shown to be nearly optimal up to the next asymptotic term for the prefix sum problem

Table 1: Learning rate decays.

| | |
|---|---|
| Exponential | $\chi_k = \beta^{\frac{k-1}{n-1}}$ |
| Polynomial | $\chi_k = \beta + (1-\beta)\dfrac{\left(\frac{n}{k}\right)^\gamma - 1}{n^\gamma - 1},\ \gamma \geq 1$ |
| Linear | $\chi_k = 1 - \frac{k-1}{n-1}(1-\beta)$ |
| Cosine | $\chi_k = \beta + \frac{1-\beta}{2}\left(1 + \cos\left(\frac{k-1}{n-1}\pi\right)\right)$ |

($\chi_k = 1$) (Henzinger et al., 2025). To further improve the bounds, we propose a learning-rate-aware factorization. To define it, let $A_\chi^{\text{Toep}}$ denote the Toeplitz matrix with $\chi_1, \ldots, \chi_n$ on its subdiagonals.

$$A_\chi^{\text{Toep}} = \begin{pmatrix} \chi_1 & 0 & 0 & \ldots & 0 \\ \chi_2 & \chi_1 & 0 & \ldots & 0 \\ \chi_3 & \chi_2 & \chi_1 & \ldots & 0 \\ \vdots & \vdots & \vdots & \ddots & \vdots \\ \chi_n & \chi_{n-1} & \chi_{n-2} & \ldots & \chi_1 \end{pmatrix} \tag{6}$$

We propose $C_\chi = (A_\chi^{\text{Toep}})^{1/2}$ as a learning-rate-aware correlation matrix. To analyze its properties, we consider the exponentially decaying learning rate $\chi_t = \beta^{\frac{t-1}{n-1}} = \alpha^{t-1}$ with $\alpha = \beta^{\frac{1}{n-1}}$. In this setting, the correlation matrix can be computed explicitly as

$$C_\alpha = \begin{pmatrix} 1 & 0 & \ldots & 0 \\ \alpha r_1 & 1 & \ldots & 0 \\ \vdots & \vdots & \ddots & \vdots \\ \alpha^{n-1} r_{n-1} & \alpha^{n-2} r_{n-2} & \ldots & 1 \end{pmatrix}, \tag{7}$$

where the coefficients are $r_j = \left| \binom{-1/2}{j} \right| = \frac{1}{4^j}\binom{2j}{j}$.

## 4 RESULTS

In this work, we derive upper and lower bounds on the MaxSE and MeanSE errors of the learning rate scheduling workload $A_\chi$ for a large class of learning rate schedulers $\chi_1, \ldots, \chi_n$. In the following theorem, we prove an upper bound based on the prefix-sum factorization $A_\chi A_1^{-1/2} \times A_1^{1/2}$.

**Theorem 1.** *Let $(\chi_t)_{t=1}^n$ be a sequence on $[\beta, 1]$ for some constant $\beta > 0$. For $n \geq 2$ we define*

$$\Delta_t = |\chi_t - \chi_{t+1}| \qquad (\text{for all } 1 \leq t \leq n-1). \tag{8}$$

*If either of the following two conditions holds ($c > 0$ an absolute constant):*

$$\Delta_t \leq \frac{c}{t(1 + \log t)} \qquad (\text{for all } 1 \leq t \leq n-1), \qquad \text{or} \qquad \sum_{t=1}^{n-1} \Delta_t^2 = o\left(\frac{\log n}{n}\right), \tag{9}$$

*then the factorization $B_\chi \times A_1^{1/2}$, where $B_\chi := A_\chi (A_1)^{-1/2}$, satisfies*

$$\text{MaxSE}(B_\chi, A_1^{1/2}) = \Theta\left( \sqrt{\log n} \cdot \sqrt{\max_{m \in [n]} \chi_m^2 \log m} \right), \tag{10}$$

$$\text{MeanSE}(B_\chi, A_1^{1/2}) = \Theta\left( \sqrt{\log n} \cdot \sqrt{\frac{1}{n}\sum_{m=1}^n \chi_m^2 \log m} \right). \tag{11}$$

The conditions assumed in Theorem 1 are satisfied for all learning rate decays presented in Table 1, more formally:

**Lemma 1.** *Every learning rate schedule $(\chi_t)_{t=1}^n$ with constant $\beta \in (0, 1/e)$ presented in Table 1 satisfies the assumptions of Theorem 1.*

Moreover, in this work we also prove general lower bounds for any learning rate schedules:

**Theorem 2.** *Let $A_\chi = A_1 D_\chi$, where $D_\chi = \mathrm{diag}(\chi_1, \ldots, \chi_n)$ with positive $\chi_t > 0$. Then*

$$\inf_{B \times C = A_\chi} MaxSE(B, C) \geq \max_{1 \leq t \leq n} \frac{1}{\pi} \left( \min_{j \leq t} \chi_j \right) \log t \tag{12}$$

$$\inf_{B \times C = A_\chi} MeanSE(B, C) \geq \max_{1 \leq t \leq n} \frac{1}{\pi} \sqrt{\frac{t}{n}} \left( \min_{j \leq t} \chi_j \right) \log t. \tag{13}$$

In particular, plugging in the exponential learning rate decay $\chi_k = \beta^{\frac{k-1}{n-1}}$ yields the following upper and lower bounds.

**Corollary 1.** *For exponential learning rate decay $\chi_k = \beta^{\frac{k-1}{n-1}}$ with $\beta \in (0, 1/e)$, the prefix-sum–based factorization $A_\chi = A_\chi(A_1)^{-1/2} \times A_1^{1/2}$ gives the following values for MaxSE and MeanSE:*

$$\mathrm{MaxSE}(B_\chi, A_1^{1/2}) = \Theta \left( \sqrt{\log n} \sqrt{\log \frac{n}{\log(1/\beta)}} \right), \tag{14}$$

$$\mathrm{MeanSE}(B_\chi, A_1^{1/2}) = \Theta \left( \frac{\log n}{\sqrt{\log(1/\beta)}} \right). \tag{15}$$

**Corollary 2.** *Suppose $\chi_k = \beta^{\frac{k-1}{n-1}}$ with $\beta \in (0, 1/e)$. Then*

$$\inf_{B \times C = A_\chi} MaxSE(B, C) = \Omega \left( \log \frac{n}{\log(1/\beta)} \right) \tag{16}$$

$$\inf_{B \times C = A_\chi} MeanSE(B, C) = \Omega \left( \frac{1}{\sqrt{\log(1/\beta)}} \log \frac{n}{\log(1/\beta)} \right). \tag{17}$$

We further improve the upper bound by considering a learning-rate–aware factorization $C = (A_\chi^{\mathrm{Toep}})^{1/2}$, which can be computed explicitly for the exponential learning rate decay $\chi_k = \beta^{\frac{k-1}{n-1}} = \alpha^{k-1}$. This yields the factorization $A_\chi = B_\alpha \times C_\alpha$, where $C_\alpha$ is defined in equation (7), and $B_\alpha$ is obtained as $A_\chi(C_\alpha)^{-1/2}$.

In Lemma 7 of Kalinin & Lampert (2024), the sensitivity of the matrix $C_\alpha$ has been computed as:

$$\|C_\alpha\|_{1 \to 2} = \mathcal{O}\left( \frac{1}{\alpha} \sqrt{\log \frac{1}{1 - \alpha^2}} \right) = \mathcal{O}\left( \sqrt{\log \frac{n}{\log(1/\beta)}} \right). \tag{18}$$

We then bound both the maximum row norm and the Frobenius norm of $B_\alpha$, which leads to the following lemma.

**Lemma 2.** *Let $\beta \in (0, 1/e)$ and $\alpha = \beta^{1/(n-1)}$. For the factorization $A_\chi = B_\alpha \times C_\alpha$,*

$$\mathrm{MaxSE}(B_\alpha, C_\alpha) = \mathcal{O}\left( \log \frac{n}{\log(1/\beta)} \right), \tag{19}$$

$$\mathrm{MeanSE}(B_\alpha, C_\alpha) = \mathcal{O}\left( \sqrt{\frac{\log n}{\log(1/\beta)}} \sqrt{\log \frac{n}{\log(1/\beta)}} \right). \tag{20}$$

This factorization achieves the **optimal rate for the MaxSE error** and, asymptotically, performs better than alternative factorizations for the MeanSE error.

We summarize the errors for the exponential learning rate decay in Table 2. In addition, we consider four alternative factorizations: the trivial factorizations $A_\chi \times I$ and $I \times A_\chi$, two prefix-sum–inspired factorizations $A_1^{1/2} \times A_1^{1/2} D$, and the square-root factorization $A_\chi^{1/2} \times A_\chi^{1/2}$. The square-root factorization is highly nontrivial to obtain since the matrix is not Toeplitz; due to space constraints, we defer the detailed discussion to Appendix B.

Table 2: Factorizations with corresponding MaxSE and MeanSE errors for exponential learning rate scheduling $\chi_t = \beta^{\frac{t-1}{n-1}}$ for $\beta \in (0, 1/e)$. The proof of the first three bounds is rather technical and can be found in Lemma 6 in Appendix. The errors for square root factorization (d) can be found in Corollary 5. Learning-rate-aware factorization (e) is computed in Lemma 2. The prefix-sum-based factorization (f) is computed in Corollary 6. The lower bounds are computed in Corollary 2 in the appendix.

| Factorization | MaxSE | MeanSE |
|---|---|---|
| (a) $A_\chi = A_1^{1/2} \times A_1^{1/2} D$ | $\Theta(\log n)$ | $\Theta(\log n)$ |
| (b) $A_\chi = A_\chi \times I$ | $\Theta\left(\sqrt{\frac{n}{\log 1/\beta}}\right)$ | $\Theta\left(\sqrt{\frac{n}{\log 1/\beta}}\right)$ |
| (c) $A_\chi = I \times A_\chi$ | $\Theta(\sqrt{n})$ | $\Theta(\sqrt{n})$ |
| (d) $A_\chi = A_\chi^{1/2} \times A_\chi^{1/2}$ | $\Omega\left(\sqrt{\log n}\sqrt{\log \frac{n}{\log 1/\beta}}\right)$ | $\Omega\left(\frac{\log n}{\sqrt{\log(1/\beta)}}\right)$ |
| (e) $A_\chi = A_\chi(A_\chi^{\text{Toep}})^{-1/2} \times (A_\chi^{\text{Toep}})^{1/2}$ | $\mathcal{O}\left(\log \frac{n}{\log 1/\beta}\right)$ | $\mathcal{O}\left(\sqrt{\frac{\log n}{\log 1/\beta}}\sqrt{\log \frac{n}{\log 1/\beta}}\right)$ |
| (f) $A_\chi = A_1 D A_1^{-1/2} \times A_1^{1/2}$ | $\Theta\left(\sqrt{\log n}\sqrt{\log \frac{n}{\log 1/\beta}}\right)$ | $\Theta\left(\frac{\log n}{\sqrt{\log 1/\beta}}\right)$ |
| Lower Bound | $\Omega\left(\log \frac{n}{\log 1/\beta}\right)$ | $\Omega\left(\frac{1}{\sqrt{\log 1/\beta}}\log\left(\frac{n}{\log 1/\beta}\right)\right)$ |

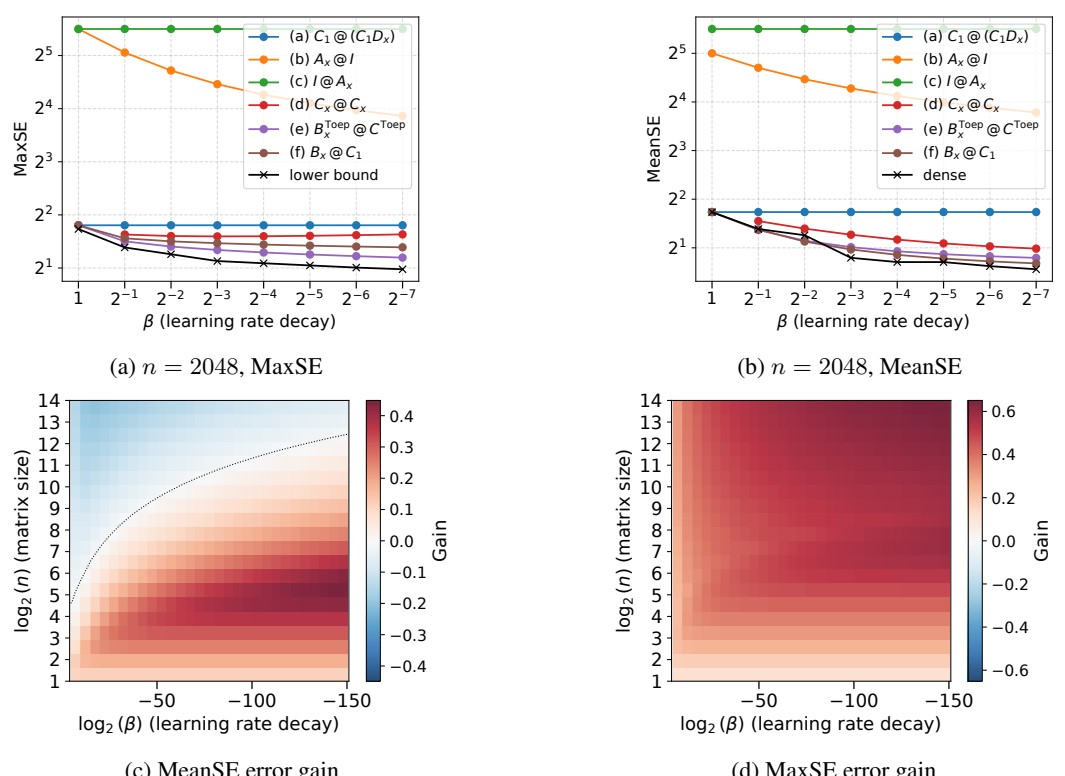

(a) $n = 2048$, MaxSE

(b) $n = 2048$, MeanSE

(c) MeanSE error gain

(d) MaxSE error gain

Figure 1: Comparison of MaxSE and MeanSE errors under an **exponentially decaying** learning rate, for the proposed factorizations (see Table 2), with fixed matrix size $n = 2048$ and varying decay $\beta$. We refer to the approximately optimal value of MeanSE computed by dense factorization (Denisov et al., 2022) as "dense." For MaxSE, we report a lower bound since no scalable and accurate solution for its optimal value is available. The bottom row compares our learning-rate aware factorization with the prefix-sum based one, validating the theoretical improvements in both MeanSE and MaxSE.

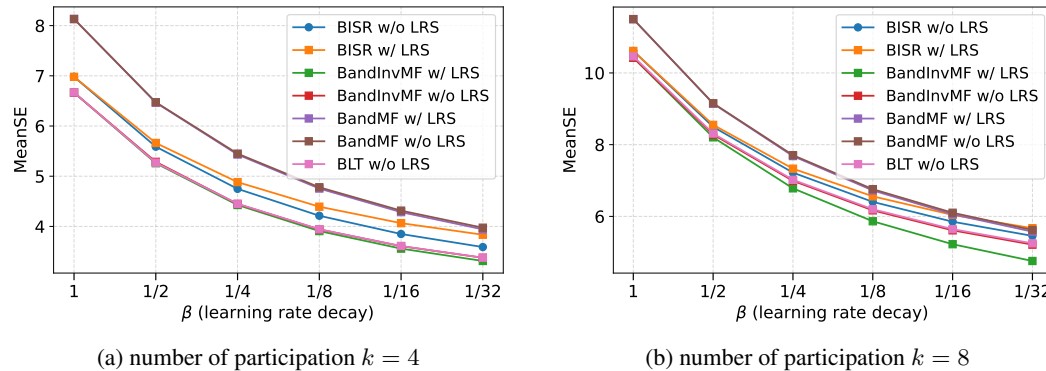

(a) number of participation $k = 4$         (b) number of participation $k = 8$

Figure 2: Multi-participation MeanSE error with matrix size $n = 2048$. Lines are computed for bandwidth $p = 64$. For the exponential workload, we observe that with a larger participation number it becomes beneficial to optimize the factorization with respect to the learning rate decay workload. However, for the considered values of $n$ and $\beta$, we do not observe any benefit from incorporating learning rate scheduling for BISR.

We then numerically compare the proposed factorizations in the single-epoch (single-participation) setting using the MaxSE and MeanSE metrics, as functions of the learning rate decay $\beta$ and the matrix size $n$ (see Figure 1 for exponential decay and Figure 4 in the appendix for other learning rate decays). As an approximation of the actual optimal value for MeanSE, we use a dense factorization (Denisov et al., 2022) implemented in jax-privacy library (Balle et al., 2025). On the plots, we refer to this approximation as "dense". For MaxSE, it is computationally infeasible to compute the exact optimal value for large matrix sizes. Therefore, we rely on the lower bound derived in Theorem 2, which we denote on the plots as "lower bound". We observe that our learning-rate-aware factorization outperforms the others in terms of MaxSE. However, for the proposed values of $n$ and $\beta$, it performs worse than the prefix sum based factorization in terms of MeanSE. To further investigate this, we plot the colormap of the gain over the prefix sum based approach (see Figure 1). In the blue regions, our method performs worse, while in the red regions it performs better. As can be seen, for any fixed $n$, sufficiently small values of $\beta$ lead to the learning-rate-aware factorization outperforming the prefix sum based approach, thereby numerically validating our theoretical findings.

## 4.1 MULTI-PARTICIPATION

Following the line of work on multi-participation matrix factorization (Choquette-Choo et al., 2023b;a; Kalinin & Lampert, 2024; McKenna, 2025; Kalinin et al., 2025), we allow each user or datapoint to participate multiple times. Without imposing any restriction on the participation pattern, the guarantees would be no stronger than those obtained via the privacy composition. To overcome this, we adopt the notion of $b$-*min separation*, which requires that the gap between any two consecutive participations of the same user be at least $b > 0$. Under this condition, each user may participate up to $k = \lceil n/b \rceil$ times. This naturally affects the definition of sensitivity, which we refine as

$$\text{sens}_{k,b}(C) = \sup_{G \sim G'} \|CG - CG'\|_F, \tag{21}$$

where $G$ and $G'$ differ in the participations of a single user, with the corresponding rows separated by at least $b$. We then generalize the notion of MeanSE error to the multi-participation setting:

$$\mathcal{E}(B, C) = \frac{1}{\sqrt{n}} \|B\|_F \cdot \text{sens}_{k,b}(C). \tag{22}$$

In this section, we establish both upper and lower bounds on the optimal value $\mathcal{E}(B, C)$ among all factorizations, for the learning-rate workload. This extends the results of Kalinin et al. (2025) on SGD with momentum and weight-decay workloads to the non-Toeplitz case. For the prefix-sum workload, it was shown that the **Banded Inverse Square Root (BISR)** factorization is asymptotically optimal in the multi-participation setting. The BISR is defined as follows: given a workload

matrix $A$, we compute the square root of its inverse, $C = A^{-1/2}$, band it to width $p$ by nullifying all elements below the $p$-th diagonal and then invert the result. The corresponding correlation matrix is denoted $C^p$. Then there exists a unique matrix $B^p$ such that $B^p C^p = A$. By using the BISR matrix corresponding to the prefix-sum workload $A_1$, we establish a general upper bound in the multi-participation setting for workloads with learning rates $A_\chi$.

**Theorem 3.** *Under the same assumptions on learning rate scheduling $\chi_t$ as in Theorem 1, the following holds.*

$$\mathcal{E}(B_\chi^p, C_1^p) = \mathcal{O}\left( \sqrt{\frac{k}{n}\left(\log p + \frac{p}{b}\right) \sum_{m=1}^{n}\left[\chi_m^2 \log\left(\min\{m,p\}\right) + \frac{1}{p}\sum_{t=p}^{m-1}\chi_t^2\right]} \right). \quad (23)$$

For exponential decay the upper bound (after optimizing over $p$) has the following form:

**Corollary 3.** *Let $\chi_t = \beta^{\frac{t-1}{n-1}}$ with $\beta \in (0, 1/e)$. Then, in multi-participation with $b$-min-separation and at most $k = \lceil \frac{n}{b} \rceil$ participations, we have for $p^* \sim b\log b$ the following optimized upper bound:*

$$\mathcal{E}(B_\chi^p, C_1^p) = \mathcal{O}\left(\frac{\sqrt{k}\log n + k}{\sqrt{\log(1/\beta)}}\right). \quad (24)$$

We prove a general lower bound for multi-participation error with arbitrary learning rate scheduling.

**Theorem 4** (Lower bound for multi-participation). *Let $A_\chi = A_1 D_\chi$, where $D_\chi = \mathrm{diag}(\chi_1, \ldots, \chi_n)$ with positive $\chi_t > 0$. Assume any factorization $A_\chi = B \times C$. Then, in multi-participation with $b$-min-separation and at most $k = \lceil \frac{n}{b} \rceil$ participations, we have*

$$\mathcal{E}(B, C) \geq \max\left\{ \max_{t \leq n}\frac{\sqrt{k}\, t\, \chi_t}{\pi\sqrt{2}n}\left(\min_{j \leq t}\chi_j\right)\log(t), \sum_{j=0}^{k-1}\chi_{1+jb}\left(1 - \frac{j}{k-1}\right)\right\}. \quad (25)$$

For the exponential learning rate decay we can simplify the lower bound in the following Corollary.

**Corollary 4.** *Let $\chi_k = \beta^{\frac{k-1}{n-1}}$ with $\beta \in (0, 1/e)$. Then Theorem 4 yields*

$$\mathcal{E}(B, C) = \Omega\left(\frac{\sqrt{k}}{\log(1/\beta)}\log\frac{n}{\log(1/\beta)} + \frac{k}{\log(1/\beta)}\right). \quad (26)$$

For the numerical comparison in the multi-participation we study several recently proposed memory-efficient factorizations. Including banded matrix factorization McKenna (2025), banded inverse factorization BandInvMF and BISR (Kalinin et al., 2025) and Buffered Linear Toeplitz (BLT) (McMahan et al., 2024). We can optimize banded and banded inverse matrices, accounting for the learning rate decay, as well as like if it was a prefix-sum workload with constant learning rate, we refer to this difference as "w/ LRS" and "w/o LRS". See the plots in the Figure 2 for the exponential decay, and Figure 5 in the Appendix for other learning rate schedulers.

## 5 EXPERIMENTS

We demonstrate the practical benefits of learning rate scheduling in Figure 3 on CIFAR-10 dataset. All experiments satisfy $(9, 10^{-5})$-DP and use a 3-block CNN trained for 10 epochs with batch size 128 and clipping norm 1. For privacy accounting, we use Poisson subsampling with PLD accounting (Koskela et al., 2021) for DP-SGD and amplification by ball-and-bins subsampling with MCMC accounting (Choquette-Choo et al., 2025) for all factorizations. Subfigure (a) shows validation accuracy across different initial learning rates, where exponential learning rate scheduling improves performance compared to DP-SGD with a fixed learning rate ($\beta = 1$). Subfigure (b) reports test accuracy using the best learning rate chosen on the validation set. All factorizations benefit substantially from scheduling, and the learning-rate–aware factorization (denoted as BISR w/ LRS) achieves even further improvements. However, optimizing the factorization with respect to learning rate workload does not lead to additional gains: while RMSE can serve as a proxy for performance,

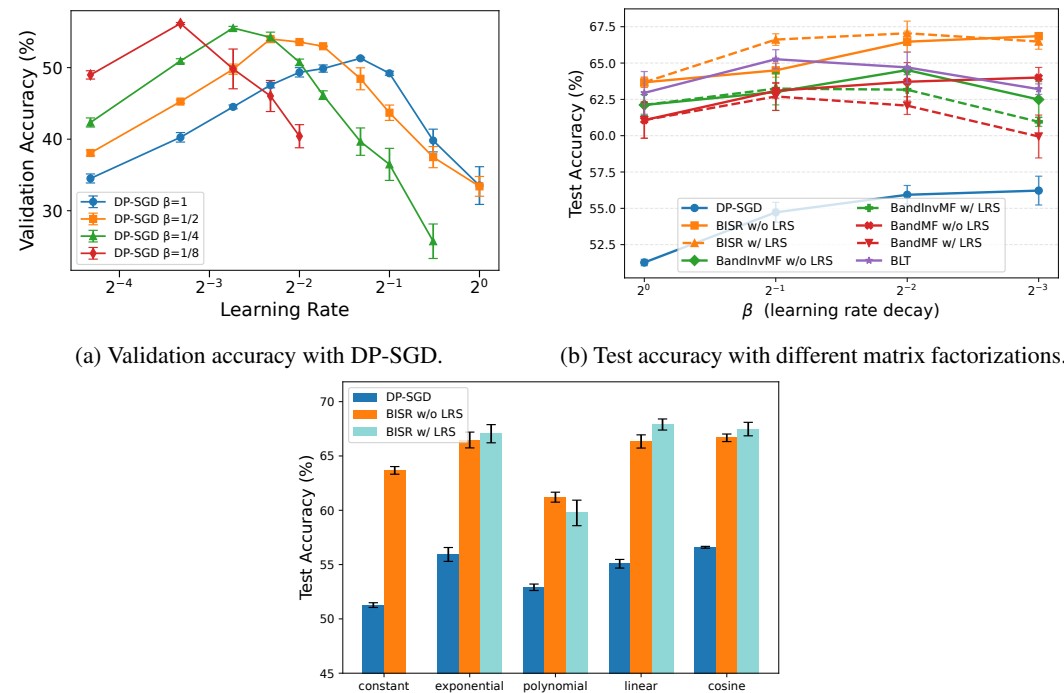

(a) Validation accuracy with DP-SGD.

(b) Test accuracy with different matrix factorizations.

(c) Test accuracy with different learning rate decays.

Figure 3: **CIFAR-10** results under $(9, 10^{-5})$-differential privacy. (a) Validation accuracy with **exponential learning rate scheduling** for different learning rates in DP-SGD. We report the points corresponding to the lowest learning rate; for example, a learning rate of $1/2$ for $\beta = 1/4$ indicates that training starts with a learning rate of $2$ and decays to $1/2$. (b) Test accuracy across different matrix factorizations with **exponential learning rate scheduling**. Training hyperparameters are provided in Table 3. (c) Test accuracy for different learning rate decays. Training hyperparameters are provided in Table 4.

it does not perfectly predict it. In practice, workload optimization increases the added noise per iteration, and this effect is not fully compensated during training due to the non-linearity introduced by large noise.

In Subfigure (c), we compare different learning rate schedulers with a constant one. We observed that learning rate scheduling improves accuracy for DP-SGD for all types. For BISR, we found that polynomial learning rate decay with $\gamma = 2$ deteriorates the quality and is perhaps not a good choice for the scheduler. The other schedulers substantially improve the accuracy of BISR. Moreover, our proposed learning-rate-aware factorization (BISR w/ LRS) further improves the quality, with the largest improvement for linear LRS, making it a suitable factorization for high-performance private training.

## 6 CONCLUSION AND FUTURE DIRECTIONS

Learning rate scheduling has been shown to improve convergence in both private and non-private machine learning. In this work, we combine learning rate scheduling with matrix factorization and propose a learning-rate-aware factorization, which in the case of exponential learning rate decay is theoretically shown to improve the error. Through numerical experiments using the MaxSE and MeanSE metrics, as well as CIFAR-10 model training, we demonstrate its benefits.

We have primarily studied learning rate decay, but similar techniques can be applied to warm-starting, where the learning rate is initially small and then gradually increased. Optimization-based approaches for matrix factorization are generally agnostic to the choice of learning rate scheduling, but adapting our learning-rate-aware factorization to this setting may pose extra challenges.

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

Figure 4: Comparison of different LR schedulers ($n = 2048$) in single participation.

# A ADDITIONAL EXPERIMENTS

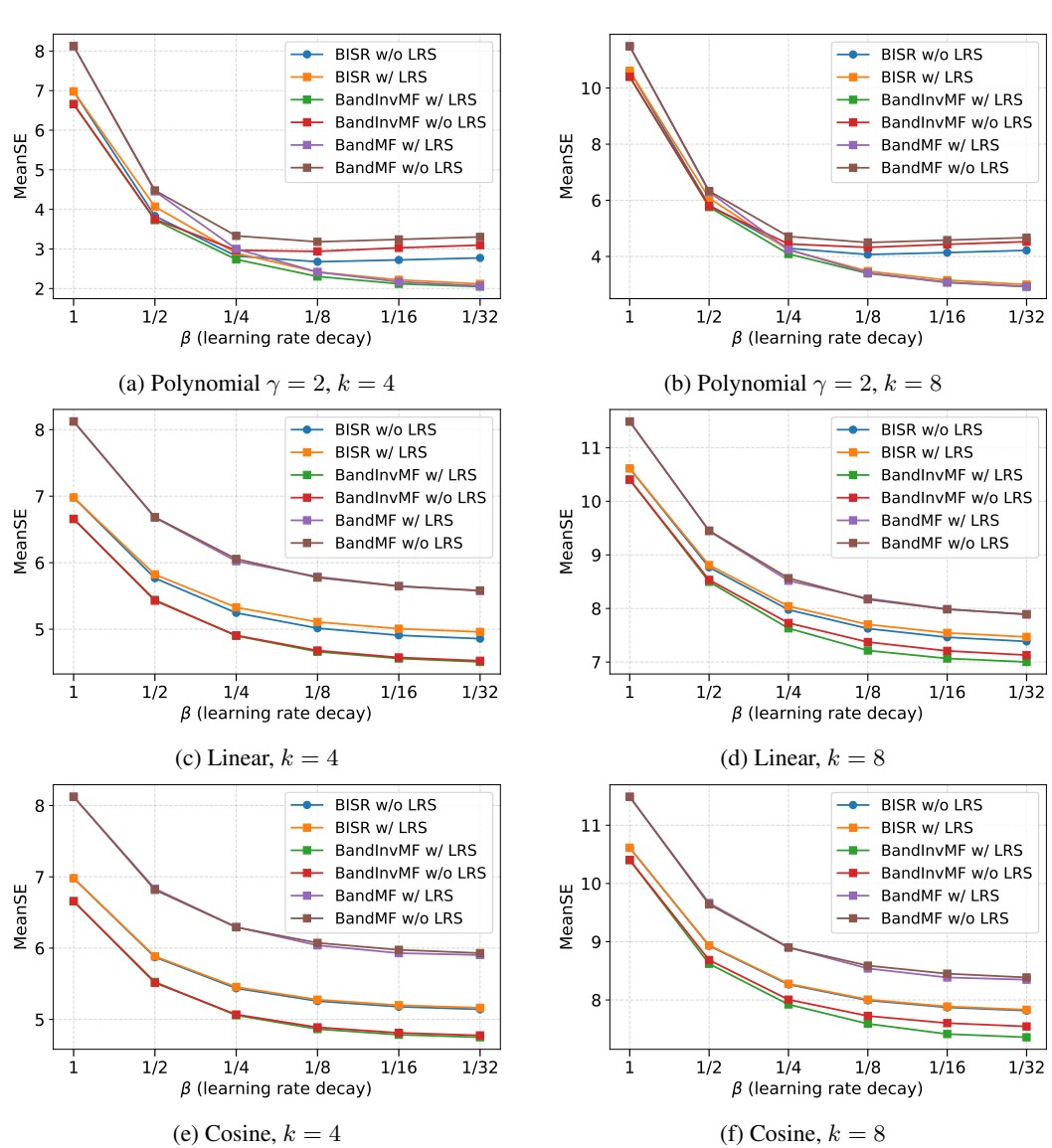

Figure 5: Multi-participation MeanSE error under different learning-rate schedulers (Polynomial $\gamma = 2$, Linear, Cosine) for $k = 4$ and $k = 8$. Matrix size $n = 1024$, bandwidth $p = 64$.

Table 3: We train four different methods for matrix optimization: DP-SGD, BISR, BandInvMF, and BandMF. Each factorization method can be computed either with a workload induced by learning rate scheduling (w/ LRS) or with a constant workload corresponding to prefix sums (w/o LRS). All experiments use clipping norm $\zeta = 1$ and batch size 128. For each method, the noise multiplier $\sigma$ is computed using a privacy accountant: Poisson accounting for DP-SGD and bins-and-balls sampling with an MCMC accountant Choquette-Choo et al. (2025) for the matrix factorization methods. Learning rates $\eta$ are tuned on a validation set separately for each method and decay setting.

| Method | $\zeta$ | BS | $p$ | $\beta = 1$ | | $\beta = \frac{1}{2}$ | | $\beta = \frac{1}{4}$ | | $\beta = \frac{1}{8}$ | |
|---|---|---|---|---|---|---|---|---|---|---|---|
| | | | | $\eta$ | $\sigma$ | $\eta$ | $\sigma$ | $\eta$ | $\sigma$ | $\eta$ | $\sigma$ |
| DP-SGD | 1 | 128 | 1 | 0.4 | 0.479 | 0.4 | 0.479 | 0.6 | 0.479 | 0.8 | 0.479 |
| BISR (w/o LRS) | 1 | 128 | 64 | 0.8 | 1.910 | 1.6 | 1.910 | 1.8 | 1.910 | 1.8 | 1.910 |
| BISR (w/ LRS) | 1 | 128 | 64 | 0.8 | 1.908 | 1.6 | 1.901 | 1.9 | 1.894 | 2.0 | 1.888 |
| BandInvMF (w/o LRS) | 1 | 128 | 64 | 1.0 | 2.597 | 1.5 | 2.597 | 1.6 | 2.597 | 1.7 | 2.597 |
| BandInvMF (w/ LRS) | 1 | 128 | 64 | 1.0 | 2.597 | 1.5 | 2.681 | 1.6 | 2.814 | 1.6 | 2.870 |
| BandMF (w/o LRS) | 1 | 128 | 64 | 0.9 | 2.921 | 1.5 | 2.921 | 1.6 | 2.921 | 1.7 | 2.921 |
| BandMF (w/ LRS) | 1 | 128 | 64 | 0.9 | 2.921 | 1.1 | 3.053 | 1.6 | 3.158 | 1.7 | 3.222 |
| BLT | 1 | 128 | 64 | 0.9 | 2.580 | 1.3 | 2.580 | 1.4 | 2.580 | 1.8 | 2.580 |

Table 4: Comparison of different learning rate schedulers for training with matrix factorization with fixed learning rate decay $\beta = \frac{1}{4}$. We evaluate DP-SGD, BISR (w/o LRS), and BISR (w/ LRS) under four learning rate decay strategies: exponential, polynomial, linear, and cosine. All experiments use clipping norm $\zeta = 1$ and batch size 128, for BISR we use bandwidth $p = 64$. Noise multipliers $\sigma$ are computed using Poisson accounting for DP-SGD and bins-and-balls MCMC accounting Choquette-Choo et al. (2025) for BISR. Learning rates $\eta$ are tuned on a validation set for each decay setting.

| Method | $\zeta$ | BS | $p$ | Exponential | | Polynomial | | Linear | | Cosine | |
|---|---|---|---|---|---|---|---|---|---|---|---|
| | | | | $\eta$ | $\sigma$ | $\eta$ | $\sigma$ | $\eta$ | $\sigma$ | $\eta$ | $\sigma$ |
| DP-SGD | 1 | 128 | 1 | 0.6 | 0.479 | 1.1 | 0.479 | 0.6 | 0.479 | 0.5 | 0.479 |
| BISR (w/o LRS) | 1 | 128 | 64 | 1.8 | 1.910 | 1.8 | 1.910 | 1.6 | 1.910 | 1.5 | 1.910 |
| BISR (w/ LRS) | 1 | 128 | 64 | 1.9 | 1.894 | 1.8 | 1.366 | 1.6 | 1.900 | 1.4 | 1.907 |

# B  SINGLE-PARTICIPATION: SQUARE ROOT OF THE WORKLOAD

As one of the baseline factorizations we propose the square root factorization

$$A_\chi = A_\chi^{1/2} \times A_\chi^{1/2}, \quad \text{where} \quad A_\chi = \begin{pmatrix} \chi_1 & 0 & 0 & \cdots & 0 \\ \chi_1 & \chi_2 & 0 & \cdots & 0 \\ \chi_1 & \chi_2 & \chi_3 & \cdots & 0 \\ \vdots & \vdots & \vdots & \ddots & \vdots \\ \chi_1 & \chi_2 & \chi_3 & \cdots & \chi_n \end{pmatrix} \tag{27}$$

In the case of exponential learning rate decay we can compute the matrix square root explicitly and tightly bound its values from below.

**Theorem 5.** *For any $n \geq 1$ and $\alpha \in (0,1)$, with learning rates $\chi_i = \alpha^{i-1}$ the following lower bound holds:*

$$(A_\chi^{1/2})_{m,l} = \alpha^{(l-1)/2} \prod_{k=1}^{m-l} \frac{1 - \alpha^{k-1/2}}{1 - \alpha^k} \geq \alpha^{(l-1)/2} \max \left\{ \left| \binom{-1/2}{n} \right|, \frac{\sqrt{1 - \alpha^2}}{\Gamma_{\alpha^2}(1/2)} \right\}, \tag{28}$$

*where $\Gamma_q(x)$ denotes the q-Gamma function, and $\lim_{\alpha \to 1^-} \Gamma_{\alpha^2}(1/2) = \Gamma(1/2) = \sqrt{\pi}$.*

We can also compute the inverse of this matrix (see Appendix, Lemma 5). Using the lower bound, we now establish the following bounds for the MaxSE and MeanSE errors under an exponentially decaying learning rate.

**Corollary 5.** *Let $\beta \in (0, 1/e)$ and $\alpha = \beta^{1/(n-1)}$. For the square-root factorization $A_\chi = A_\chi^{1/2} A_\chi^{1/2}$, we have*

$$\text{MaxSE}(A_\chi^{1/2}, A_\chi^{1/2}) = \Omega \left( \sqrt{\log n} \sqrt{\log \frac{n}{\log(1/\beta)}} \right), \tag{29}$$

$$\text{MeanSE}(A_\chi^{1/2}, A_\chi^{1/2}) = \Omega \left( \frac{\log n}{\sqrt{\log(1/\beta)}} \right). \tag{30}$$

We prove these statements next, beginning with necessary lemmas.

**Lemma 3.** *For a specific choice of the learning rate coefficients $\chi_i = \alpha^{2i}$ with $\alpha \in (0,1)$, we have:*

$$(A_\chi^{1/2})_{m,l} = \alpha^l \prod_{k=1}^{m-l} \frac{1 - \alpha^{2k-1}}{1 - \alpha^{2k}} \tag{31}$$

*Proof.* To prove that the coefficients of the square root have the proposed form, we need to show that the square of this matrix is equal to the original one. That is, for all $1 \leq l \leq m \leq n$, we show that:

$$\sum_{j=l}^{m} \alpha^j \prod_{k=1}^{m-j} \frac{1 - \alpha^{2k-1}}{1 - \alpha^{2k}} \cdot \alpha^l \prod_{k=1}^{j-l} \frac{1 - \alpha^{2k-1}}{1 - \alpha^{2k}} = \alpha^{2l} \tag{32}$$

or equivalently,

$$\sum_{j=0}^{m-l} \alpha^j \prod_{k=1}^{m-l-j} \frac{1 - \alpha^{2k-1}}{1 - \alpha^{2k}} \prod_{k=1}^{j} \frac{1 - \alpha^{2k-1}}{1 - \alpha^{2k}} = 1, \tag{33}$$

which is a convolution of the sequences $a_j$ and $a_j \alpha^j$, where

$$a_j = \prod_{k=1}^{j} \frac{1 - \alpha^{2k-1}}{1 - \alpha^{2k}} = \frac{(\alpha; \alpha^2)_j}{(\alpha^2; \alpha^2)_j}, \tag{34}$$

and $(a; q)_n$ denotes the $q$-Pochhammer symbol, given by $\prod_{k=0}^{n-1}(1 - aq^k)$. We will prove the identity using generating functions. First, we find the generating function of $a_j$:

$$f(x) = \sum_{j=0}^{\infty} a_j x^j = \sum_{j=0}^{\infty} \frac{(\alpha; \alpha^2)_j}{(\alpha^2; \alpha^2)_j} x^j = \frac{(\alpha x; \alpha^2)_\infty}{(x; \alpha^2)_\infty}, \tag{35}$$

where the last equality follows from the $q$-binomial theorem. Therefore, the generating function of the convolution of $a_j$ and $a_j \alpha^j$ is:

$$f(x)f(\alpha x) = \frac{(\alpha x; \alpha^2)_\infty}{(x; \alpha^2)_\infty} \cdot \frac{(\alpha^2 x; \alpha^2)_\infty}{(\alpha x; \alpha^2)_\infty} = \frac{(\alpha^2 x; \alpha^2)_\infty}{(x; \alpha^2)_\infty} = \prod_{n=0}^{\infty} \frac{1 - x\alpha^{2n+2}}{1 - x\alpha^{2n}} = \frac{1}{1-x}, \tag{36}$$

as the product telescopes, yielding the generating function of the unit sequence $(1, 1, 1, \dots)$, thus concluding the proof. $\qquad\square$

**Lemma 4.** *For any $n \geq 1$ and $\alpha \in (0, 1)$, the following lower bound holds:*

$$\prod_{k=1}^{n} \frac{1 - \alpha^{2k-1}}{1 - \alpha^{2k}} \geq \max \left\{ \left| \binom{-1/2}{n} \right|, \frac{\sqrt{1 - \alpha^2}}{\Gamma_{\alpha^2}(1/2)} \right\}, \tag{37}$$

*where $\Gamma_q(x)$ denotes the $q$-Gamma function, and $\lim_{\alpha \to 1^-} \Gamma_{\alpha^2}(1/2) = \Gamma(1/2) = \sqrt{\pi}$.*

*Proof.* First, we show that $f_n(\alpha) = \prod_{k=1}^{n} \frac{1 - \alpha^{2k-1}}{1 - \alpha^{2k}}$ is a decreasing function of $\alpha$. Therefore,

$$f_n(\alpha) \geq f_n(1) = \prod_{k=1}^{n} \frac{2k-1}{2k} = \left| \binom{-1/2}{n} \right|. \tag{38}$$

To prove this, we observe that each individual term is a decreasing function of $\alpha$:

$$\frac{1 - \alpha^{2k-1}}{1 - \alpha^{2k}} = 1 - \frac{\alpha^{2k-1} - \alpha^{2k}}{1 - \alpha^{2k}} = 1 - \frac{\alpha^{-1} - 1}{\alpha^{-2k} - 1} = 1 - \frac{1}{1 + \alpha^{-1} + \cdots + \alpha^{-(2k-1)}}. \tag{39}$$

For the second part of the inequality, we show that

$$f_n(\alpha) \geq f_\infty(\alpha) = \prod_{k=1}^{\infty} \frac{1 - \alpha^{2k-1}}{1 - \alpha^{2k}} = \frac{(\alpha; \alpha^2)_\infty}{(\alpha^2; \alpha^2)_\infty} = \frac{\sqrt{1 - \alpha^2}}{\Gamma_{\alpha^2}(1/2)}, \tag{40}$$

where the inequality holds because each term of the product is less than 1, the infinite product converges, and the $q$-Gamma function is defined by

$$\Gamma_q(x) = (1 - q)^{1-x} \frac{(q; q)_\infty}{(q^x; q)_\infty}. \tag{41}$$

This concludes the proof. $\qquad\square$

*Proof of Theorem 5.* The proof follows from combining Lemma 3 for the equality and Lemma 4 for the lower bound. For convenience, we considered $\chi_i = \alpha^{2i}$ in those lemmas. To achieve $\alpha^{i-1}$, we first divide the square root matrix by $\alpha$ so that we start from learning rates of 1 rather than $\alpha^2$. Then, we replace $\alpha$ with $\alpha^{1/2}$, which concludes the proof. $\qquad\square$

*Proof of Corollary 5.* To use Lemma 3 and Lemma 4, we need to adjust the choice of $\alpha$, as previous lemmas consider $\chi_k = \alpha^{2k}$ while here $\chi_k = \alpha^{k-1}$. This gives

$$(A_\chi^{1/2})_{m,l} \geq \alpha^{(l-1)/2} r_{m-l}. \tag{42}$$

Thus the maximum column norm of $A_\chi^{1/2}$ is at least the norm of its first column, which in turn is at least the maximum column norm of $A_1^{1/2}$; the latter is $\Theta(\log n)$.

For the $m$-th row-sum of squares,

$$\sum_{l=1}^{m}(A_\chi^{1/2})_{m,l}^2 \geq \sum_{l=1}^{m}\alpha^{l-1}r_{m-l}^2 \geq \frac{\alpha^m}{\pi}\sum_{l=1}^{m}\frac{1}{l\,\alpha^l} \geq \frac{\alpha^m}{\pi}\log m.$$

**MaxSE.** Taking the maximum over $m$ and applying Lemma 10 yields

$$\max_{1\leq m\leq n}\sum_{l=1}^{m}(A_\chi^{1/2})_{m,l}^2 = \Omega\left(\log\frac{n}{\log(1/\beta)}\right),$$

so the maximum row norm is $\Omega\big(\sqrt{\log\frac{n}{\log(1/\beta)}}\big)$. Multiplying by the maximum column norm $\Omega(\log n)$ gives the first bound.

**MeanSE.** Averaging over $m$ and using Lemma 11,

$$\frac{1}{n}\sum_{m=1}^{n}\sum_{l=1}^{m}(A_\chi^{1/2})_{m,l}^2 = \Theta\left(\frac{\log n}{\log(1/\beta)}\right),$$

so the average row norm is $\Omega\big(\sqrt{\frac{\log n}{\log(1/\beta)}}\big)$. Multiplying by the maximum column norm $\Omega(\log n)$ gives the second bound. $\qquad\square$

We also give the following lemma on the inverse of $A_\chi^{1/2}$.

**Lemma 5.** *The inverse matrix $A_\chi^{-1/2}$ for a specific choice of the learning rate coefficients $\chi_i = \alpha^{2i}$ with $\alpha \in (0,1)$ has the following form:*

$$(A_\chi^{-1/2})_{m,l} = \frac{-(1-\alpha)}{\alpha^{l+1}(1-\alpha^{2(m-l)-1})}\prod_{k=1}^{m-l}\frac{1-\alpha^{2k-1}}{1-\alpha^{2k}}. \tag{43}$$

*Proof.* To prove that $A_\chi^{-1/2}$ corresponds to the inverse square root matrix, we will show that its product with the square root matrix $A_\chi^{1/2}$ yields the identity matrix:

$$\sum_{j=l}^{m}(A_\chi^{1/2})_{m,j}(A_\chi^{-1/2})_{j,l} = \sum_{j=l}^{m}\alpha^j\left[\prod_{k=1}^{m-j}\frac{1-\alpha^{2k-1}}{1-\alpha^{2k}}\right]\cdot\frac{\alpha-1}{\alpha^{l+1}(1-\alpha^{2(j-l)-1})}\left[\prod_{k=1}^{j-l}\frac{1-\alpha^{2k-1}}{1-\alpha^{2k}}\right] = \mathbb{1}_{l=m}. \tag{44}$$

This is equivalent to proving the following identity:

$$\sum_{j=0}^{m-l}\alpha^j\left[\prod_{k=1}^{m-l-j}\frac{1-\alpha^{2k-1}}{1-\alpha^{2k}}\right]\cdot\frac{\alpha-1}{\alpha(1-\alpha^{2j-1})}\left[\prod_{k=1}^{j}\frac{1-\alpha^{2k-1}}{1-\alpha^{2k}}\right] = \mathbb{1}_{l=m}. \tag{45}$$

This can be interpreted as a convolution of two sequences $a_j$ and $\alpha^j b_j$, defined as follows:

$$a_j = \prod_{k=1}^{j}\frac{1-\alpha^{2k-1}}{1-\alpha^{2k}} = \frac{(\alpha;\alpha^2)_j}{(\alpha^2;\alpha^2)_j}, \qquad b_j = \frac{\alpha-1}{\alpha(1-\alpha^{2j-1})}\prod_{k=1}^{j}\frac{1-\alpha^{2k-1}}{1-\alpha^{2k}} = \frac{(1/\alpha;\alpha^2)_j}{(\alpha^2;\alpha^2)_j}, \tag{46}$$

where $(a;q)_n$ denotes the $q$-Pochhammer symbol, given by $\prod_{k=0}^{n-1}(1-aq^k)$.

Analogously to Lemma 3, we will prove the identity via generating functions. Since the generating function $f(x)$ of $a_j$ is already known, it remains to find the generating function of $b_j$:

$$g(x) = \sum_{j=0}^{\infty}x^j b_j = \sum_{j=0}^{\infty}x^j\frac{(1/\alpha;\alpha^2)_j}{(\alpha^2;\alpha^2)_j} = \frac{(x/\alpha;\alpha^2)_\infty}{(x;\alpha^2)_\infty}. \tag{47}$$

Therefore, the generating function of the convolution of $a_j$ and $\alpha^j b_j$ is given by the product of the generating functions $f(x)$ and $g(\alpha x)$, resulting in:

$$f(x)g(\alpha x) = \frac{(\alpha x; \alpha^2)_\infty}{(x; \alpha^2)_\infty} \cdot \frac{(x; \alpha^2)_\infty}{(\alpha x; \alpha^2)_\infty} = 1, \tag{48}$$

which concludes the proof. □

## C SINGLE PARTICIPATION: NAIVE FACTORIZATIONS

**Lemma 6.**

(a) $MaxSE(A_1^{1/2}, A_1^{1/2}D) = \Theta(\log n)$      $MeanSE(A_1^{1/2}, A_1^{1/2}D) = \Theta(\log n)$

(b) $MaxSE(A_\chi, I) = \Theta\left(\sqrt{\frac{n}{\log 1/\beta}}\right)$      $MeanSE(A_\chi, I) = \Theta\left(\sqrt{\frac{n}{\log 1/\beta}}\right)$

(c) $MaxSE(I, A_\chi) = \Theta(\log n)$      $MeanSE(I, A_\chi) = \Theta(\log n)$

*Proof.* **(a)** Since $\chi_1 = 1$ and all other $\chi_t \leq 1$, the maximum column norm is still achieved in the first column and is exactly the same as that of $A_1^{1/2}$. Thus,

$$\mathrm{MaxSE}(A_1^{1/2}, A_1^{1/2}D) = \mathrm{MaxSE}(A_1^{1/2}, A_1^{1/2}) = \Theta(\log n),$$

$$\mathrm{MeanSE}(A_1^{1/2}, A_1^{1/2}D) = \mathrm{MeanSE}(A_1^{1/2}, A_1^{1/2}) = \Theta(\log n),$$

which follows from the analysis of the prefix-sum square root factorization by Henzinger et al. (2024).

**(b)** The maximum column norm of $I$ is 1. The maximum row norm of $A_\chi$ is

$$\sqrt{\sum_{k=1}^{n} \chi_k^2} = \sqrt{\sum_{k=0}^{n-1} \beta^{\frac{2k}{n-1}}} = \sqrt{\frac{1 - \beta^{\frac{2n}{n-1}}}{1 - \beta^{\frac{2}{n-1}}}} = \Theta\left(\sqrt{\frac{n}{\log(1/\beta)}}\right). \tag{49}$$

The normalized Frobenius norm $\frac{1}{\sqrt{n}}\|A_\chi\|_F$ is

$$\frac{1}{\sqrt{n}}\|A_\chi\|_F = \frac{1}{\sqrt{n}}\sqrt{\sum_{k=1}^{n}(n+1-k)\chi_k^2} = \frac{1}{\sqrt{n}}\sqrt{\sum_{k=1}^{n}(n+1-k)\beta^{\frac{2(k-1)}{n-1}}}$$

$$= \frac{1}{\sqrt{n}}\sqrt{\sum_{k=0}^{n-1}(n-k)\beta^{\frac{2k}{n-1}}} = \sqrt{\frac{\alpha^{2(n+1)} - \alpha^2(n+1) + n}{n(1-\alpha^2)^2}},$$

where $\alpha = \beta^{\frac{1}{n-1}}$. Hence $1 - \alpha^2 \sim \frac{2\log(1/\beta)}{n}$ and $\alpha^{2n} \sim \beta^2$, which results in

$$\mathrm{MeanSE}(A_\chi, I) = \frac{1}{\sqrt{n}}\|A_\chi\|_F = \Theta\left(\sqrt{\frac{1}{1-\alpha^2}}\right) = \Theta\left(\sqrt{\frac{n}{\log(1/\beta)}}\right).$$

**(c)** The maximum row norm of $I$ is 1, as is its normalized Frobenius norm. The maximum column norm of $A_\chi$ is attained in the first column and is exactly $\sqrt{n}$, which concludes the proof. □

## D SINGLE-PARTICIPATION: LEARNING-RATE-AWARE TOEPLITZ SQUARE-ROOT

**Lemma 2.** *Let $\beta \in (0, 1/e)$ and $\alpha = \beta^{1/(n-1)}$. For the factorization $A_\chi = B_\alpha \times C_\alpha$,*

$$\mathrm{MaxSE}(B_\alpha, C_\alpha) = \mathcal{O}\left(\log \frac{n}{\log(1/\beta)}\right), \tag{19}$$

$$\mathrm{MeanSE}(B_\alpha, C_\alpha) = \mathcal{O}\left(\sqrt{\frac{\log n}{\log(1/\beta)}} \sqrt{\log \frac{n}{\log(1/\beta)}}\right). \tag{20}$$

*Proof.* For the exponential learning rate $\chi_k = \beta^{\frac{k-1}{n-1}} = \alpha^{k-1}$, consider the factorization

$$A_\chi = A_\chi C_\alpha^{-1} \times C_\alpha = B_\alpha \times C_\alpha,$$

with

$$C_\alpha = \begin{pmatrix} 1 & 0 & \dots & 0 \\ \alpha r_1 & 1 & \dots & 0 \\ \vdots & \vdots & \ddots & \vdots \\ \alpha^{n-1} r_{n-1} & \alpha^{n-2} r_{n-2} & \dots & 1 \end{pmatrix}, \qquad C_\alpha^{-1} = \begin{pmatrix} 1 & 0 & \dots & 0 \\ \alpha \tilde{r}_1 & 1 & \dots & 0 \\ \vdots & \vdots & \ddots & \vdots \\ \alpha^{n-1} \tilde{r}_{n-1} & \alpha^{n-2} \tilde{r}_{n-2} & \dots & 1 \end{pmatrix},$$
(50)

where $(r_k)$ and $(\tilde{r}_k)$ are the coefficients of $A_1^{1/2}$ and $A_1^{-1/2}$, respectively. By equation (18),

$$\|C_\alpha\|_{1\to 2} = \mathcal{O}\Big(\frac{1}{\alpha}\sqrt{\log \frac{1}{1-\alpha^2}}\Big) = \mathcal{O}\Big(\sqrt{\log \frac{n}{\log(1/\beta)}}\Big).$$
(51)

The corresponding left matrix is $B_\alpha$, which can be computed as

$$(B_\alpha)_{m,l} = \alpha^l \sum_{t=0}^{m-l} \tilde{r}_t \alpha^{2t}.$$

Then, applying summation by parts:

$$(B_\alpha)_{m,l} = \alpha^{2m-l} r_{m-l} + \alpha^l (1-\alpha^2) \sum_{t=0}^{m-l-1} \alpha^{2t} r_t.$$
(52)

Using $(a+b)^2 \le 2a^2 + 2b^2$ and summing over $l$,

$$\sum_{l=1}^m (B_\alpha)_{m,l}^2 \le 2 \sum_{l=1}^m \alpha^{4m-2l} r_{m-l}^2 + 2(1-\alpha^2)^2 \sum_{l=1}^m \alpha^{2l} \Big( \sum_{t=0}^{m-l-1} \alpha^{2t} r_t \Big)^2.$$
(53)

*First term.* Since $\sum_{u=0}^{m-1} r_u^2 \le 1 + \frac{1}{\pi} + \frac{1}{\pi}\log(m-1)$,

$$2 \sum_{l=1}^m \alpha^{4m-2l} r_{m-l}^2 = 2\alpha^{2m} \sum_{u=0}^{m-1} \alpha^{2u} r_u^2 \le \frac{2}{\pi} \alpha^{2m} \log m + \mathcal{O}(1).$$
(54)

*Second term.* With $r_0 = 1$ and $r_t \le \frac{1}{\sqrt{\pi t}}$ for $t \ge 1$,

$$\sum_{t=0}^{m-l-1} \alpha^{2t} r_t \le 1 + \frac{1}{\sqrt{\pi}} \sum_{t=1}^{\infty} \frac{\alpha^{2t}}{\sqrt{t}}, \qquad \sum_{l=1}^m \alpha^{2l} \le \frac{\alpha^2}{1-\alpha^2},$$

and

$$\sum_{t=1}^{\infty} \frac{\alpha^{2t}}{\sqrt{t}} \le \int_0^{\infty} \frac{e^{2x \log \alpha}}{\sqrt{x}} \, dx = \frac{\sqrt{\pi}}{\sqrt{2\log(1/\alpha)}}.$$
(55)

Hence

$$2(1-\alpha^2)^2 \sum_{l=1}^m \alpha^{2l} \Big( \sum_{t=0}^{m-l-1} \alpha^{2t} r_t \Big)^2 \le 2\alpha^2 (1-\alpha^2) \Big( 1 + \frac{1}{\sqrt{2\log(1/\alpha)}} \Big)^2 = \mathcal{O}(1).$$
(56)

Combining (54)–(56),

$$\sum_{l=1}^m (B_\alpha)_{m,l}^2 \le \frac{2}{\pi} \alpha^{2m} \log m + \mathcal{O}(1).$$
(57)

**MaxSE.** From (57),

$$\|B_\alpha\|_{2\to\infty}^2 = \max_m \sum_{l=1}^m (B_\alpha)_{m,l}^2 \le \max_m \Big( \frac{2}{\pi} \alpha^{2m} \log m + \mathcal{O}(1) \Big) = \mathcal{O}\Big( \log \frac{n}{\log(1/\beta)} \Big)$$

by Lemma 10 with $\alpha \mapsto \alpha^2$. Multiplying by $\|C_\alpha\|_{1\to2} = \mathcal{O}\big(\frac{1}{\alpha}\sqrt{\log\frac{1}{1-\alpha^2}}\big)$ (Kalinin & Lampert, 2024, Lemma 7) gives the stated bound.

**MeanSE.** From (57),

$$\frac{1}{n}\|B_\alpha\|_F^2 = \frac{1}{n}\sum_{m=1}^{n}\sum_{l=1}^{m}(B_\alpha)_{m,l}^2 \leq \frac{2}{\pi}\cdot\frac{1}{n}\sum_{m=1}^{n}\alpha^{2m}\log m + \mathcal{O}(1) = \mathcal{O}\left(\frac{\log n}{\log(1/\beta)}\right)$$

by Lemma 11 with $\alpha \mapsto \alpha^2$. Thus $\frac{1}{\sqrt{n}}\|B_\alpha\|_F = \mathcal{O}\big(\sqrt{\frac{\log n}{\log(1/\beta)}}\big)$, and multiplying by $\|C_\alpha\|_{1\to2}$ (as above) yields the claimed bound. $\qquad\square$

## E  SINGLE-PARTICIPATION: PREFIX-SUM FACTORIZATION

**Lemma 7.** *Let $(\chi_t)_{t=1}^{n}$ be a positive sequence taken from $[\beta,\infty)$ where $\beta > 0$ is a constant, and*

$$Q = \sum_{l=1}^{n-1}\left(\sum_{t=0}^{n-l-1}|\chi_{l+t} - \chi_{l+t+1}|r_t\right)^2 = o(\log n)\,.$$

*Then*

$$\mathrm{MaxSE}(B_\chi, A_1^{1/2}) = \Theta\left(\sqrt{\log n}\cdot\sqrt{\max_{m\in[n]}\chi_m^2\log m}\right),$$

$$\mathrm{MeanSE}(B_\chi, A_1^{1/2}) = \Theta\left(\sqrt{\log n}\cdot\sqrt{\frac{1}{n}\sum_{m=1}^{n}\chi_m^2\log m}\right)\,.$$

*Proof.* We have that

$$(B_\chi)_{m,l} = \chi_l + \sum_{t=1}^{m-l}\tilde{r}_t\chi_{t+l}$$

where $\tilde{r}_t = \frac{-r_t}{2t-1}$ are the coefficients of $A_1^{-1/2}$. Applying summation by parts (also known as Abel's transformation), we obtain:

$$(B_\chi)_{m,l} = \chi_m\sum_{t=0}^{m-l}\tilde{r}_t - \sum_{t=0}^{m-l-1}(\chi_{l+t+1} - \chi_{l+t})\sum_{j=0}^{t}\tilde{r}_j$$

$$= \chi_m r_{m-l} + \sum_{t=0}^{m-l-1}(\chi_{l+t} - \chi_{l+t+1})r_t\,. \tag{58}$$

Defining $\Delta_k = |\chi_k - \chi_{k+1}|$, and using $(a+b)^2 \leq 2a^2 + 2b^2$, we get for its squared row sum:

$$\sum_{l=1}^{m}(B_\chi)_{m,l}^2 \leq 2\sum_{l=1}^{m}\chi_m^2 r_{m-l}^2 + 2\underbrace{\sum_{l=1}^{m-1}\left(\sum_{t=0}^{m-l-1}\Delta_{l+t}r_t\right)^2}_{Q_m}$$

Similarly, using the inequality $(a+b)^2 \geq \frac{1}{2}a^2 - b^2$:

$$\sum_{l=1}^{m}(B_\chi)_{m,l}^2 \geq \frac{1}{2}\chi_m^2\sum_{l=1}^{m}r_{m-l}^2 - Q_m$$

Using the bounds $\frac{1}{\sqrt{\pi(t+1)}} \leq r_t \leq \frac{1}{\sqrt{\pi t}}$, we arrive at

$$\frac{1}{2\pi}\chi_m^2\log m + \mathcal{O}(1) - Q_m \leq \sum_{l=1}^{m}(B_\chi)_{m,l}^2 \leq \frac{2}{\pi}\chi_m^2\log m + \mathcal{O}(1) + 2Q_m\,.$$

As $Q_m \leq Q$ (observe that $Q_m$ is a truncated sum of positive summands), we can write

$$\sum_{l=1}^{m} (B_\chi)_{m,l}^2 = \Theta\left(\chi_m^2 \log m\right) \pm \mathcal{O}(Q).$$

For the requisite norms, we get:

$$\|B_\chi\|_{2\to\infty}^2 = \max_{m\in[n]} \sum_{l=1}^{m} (B_\chi)_{m,l}^2 = \Theta\left(\underbrace{\max_{1\leq m\leq n} \chi_m^2 \log m}_{\Theta(\log n)}\right) \pm \mathcal{O}(Q) = \Theta\left(\max_{1\leq m\leq n} \chi_m^2 \log m\right)$$

$$\frac{1}{n}\|B_\chi\|_F^2 = \frac{1}{n}\sum_{m=1}^{n}\sum_{l=1}^{m} (B_\chi)_{m,l}^2 = \Theta\left(\underbrace{\frac{1}{n}\sum_{m=1}^{n} \chi_m^2 \log m}_{\Theta(\log n)}\right) \pm \mathcal{O}(Q) = \Theta\left(\frac{1}{n}\sum_{m=1}^{n} \chi_m^2 \log m\right)$$

as $Q = o(\log n)$ by assumption. The final statement is derived from taking the square-root of the requisite norm and multiplying by $\|A_1^{1/2}\|_{1\to2} = \Theta(\log n)$. □

**Lemma 8.** *Let $(\chi_t)_{t=1}^{n}$ be a positive sequence. Fix $n \geq 2$ and define $(\Delta_t)_{t=1}^{n-1}$ via*

$$\Delta_t = |\chi_t - \chi_{t+1}| \qquad (\text{for all } 1 \leq t \leq n-1).$$

*Then*

$$Q = \sum_{l=1}^{n-1}\left(\sum_{t=0}^{n-l-1} |\chi_{l+t} - \chi_{l+t+1}| r_t\right)^2 = \mathcal{O}\left(n\sum_{k=1}^{n-1} \Delta_k^2\right).$$

*Proof.* Define the two sequences $(a_t)_{t\in\mathbb{Z}}, (b_t)_{t\in\mathbb{Z}}$ via

$$a_t = \begin{cases} \Delta_t & \text{for } 1 \leq t \leq n-1 \\ 0 & \text{otherwise} \end{cases} \qquad b_t = \begin{cases} r_{-t} & \text{for } 2-n \leq t \leq 0 \\ 0 & \text{otherwise} \end{cases}$$

Note that $a, b$ and $a * b$ are all in $\ell^p(\mathbb{Z})$ as they are zero-padded finite sequences, moreover

$$(a * b)_l = \sum_{t=-\infty}^{\infty} a_t b_{l-t} = \sum_{t=1}^{n-1} \Delta_t b_{l-t} = \sum_{t=l}^{n-1} \Delta_t r_{t-l} = \sum_{t=0}^{n-l-1} \Delta_{l+t} r_t.$$

We can thus write

$$Q \leq \sum_{l=-\infty}^{\infty} (a * b)_l^2 = \|a * b\|_2^2 \leq \|a\|_2^2 \cdot \|b\|_1^2 = \mathcal{O}\left(n\sum_{t=1}^{n-1} \Delta_t^2\right)$$

where the second inequality is Young's convolution inequality, and the last step uses $\sum_{t=0}^{n-2} r_t = \mathcal{O}(\sqrt{n})$. □

**Lemma 9.** *Fix $n \geq 2$, and let $(\chi_t)_{t=1}^{n}$ be a positive sequence satisfying*

$$\Delta_k = |\chi_k - \chi_{k+1}| \leq \frac{C}{k(1 + \log k)}$$

*for some absolute constant $C > 0$. Then*

$$Q = \sum_{l=1}^{n-1}\left(\sum_{t=0}^{n-l-1} |\chi_{l+t} - \chi_{l+t+1}| r_t\right)^2 = \mathcal{O}(1).$$

*Proof.* We consider the (coarse) upper bound $r_t \leq \frac{1}{\sqrt{\pi t}} < \frac{1}{\sqrt{t+1}}$, and start manipulating $Q$:

$$Q = \sum_{l=1}^{n-1} \left( \sum_{t=0}^{n-l-1} \Delta_{l+t} r_t \right)^2 \leq \sum_{l=1}^{n-1} \left( \sum_{t=0}^{n-l-1} \frac{\Delta_{l+t}}{\sqrt{t+1}} \right)^2 = \sum_{l=1}^{n-1} \left( \sum_{t=l}^{n-1} \frac{\Delta_t}{\sqrt{t-l+1}} \right)^2$$

$$= \sum_{l=1}^{n-1} \sum_{j=l}^{n-1} \sum_{k=l}^{n-1} \frac{\Delta_j \Delta_k}{\sqrt{j-l+1}\sqrt{k-l+1}}$$

$$= \sum_{j=1}^{n-1} \sum_{k=1}^{n-1} \Delta_j \Delta_k \sum_{l=1}^{\min\{j,k\}} \frac{1}{\sqrt{j-l+1}\sqrt{k-l+1}} = S$$

Our task is now reduced to showing that $S = \mathcal{O}(1)$. For $1 \leq j \leq k$, define $H(j,k) = \sum_{r=1}^{j} \frac{1}{\sqrt{r}\sqrt{r+k-j}}$. We can now write

$$S = \sum_{k=1}^{n-1} \Delta_k^2 H(k,k) + 2 \sum_{k=1}^{n-1} \sum_{j<k} \Delta_j \Delta_k H(j,k)$$

$$= \underbrace{\sum_{k=1}^{n-1} \Delta_k^2 H(k,k)}_{D} + 2\underbrace{\sum_{k=1}^{n-1} \sum_{j=1}^{\lceil k/2 \rceil} \Delta_j \Delta_k H(j,k)}_{F} + 2\underbrace{\sum_{k=1}^{n-1} \sum_{j=\lceil k/2 \rceil+1}^{k-1} \Delta_j \Delta_k H(j,k)}_{N}$$

$$= D + 2F + 2N \,.$$

We bound each separately.

**Bounding $D$.** For $D$, we have that $H(k,k) = \sum_{r=1}^{k} r^{-1} = \mathcal{O}(1 + \log k)$, and so

$$D = \sum_{k=1}^{n-1} \Delta_k^2 H(k,k) \leq \sum_{k=1}^{n-1} \frac{C'}{k^2(1 + \log k)} = \mathcal{O}(1) \,.$$

**Bounding $F$.** We have that $k - j \geq k - \lceil k/2 \rceil \geq k/2$, and so

$$H(j,k) = \sum_{r=1}^{j} \frac{1}{\sqrt{r}\sqrt{r+k-j}} \leq \sqrt{\frac{2}{k}} \sum_{r=1}^{j} \frac{1}{\sqrt{r}} \leq 2\sqrt{2}\sqrt{\frac{j}{k}} \,.$$

Plugging into our expression for $F$:

$$F = \sum_{k=1}^{n-1} \sum_{j=1}^{\lceil k/2 \rceil} \Delta_j \Delta_k H(j,k) \leq 2\sqrt{2} C^2 \sum_{k=1}^{n-1} \frac{1}{k^{3/2}(1 + \log k)} \underbrace{\sum_{j=1}^{\lceil k/2 \rceil} \frac{1}{\sqrt{j}(1 + \log j)}}_{T(\lceil k/2 \rceil)} \,.$$

We will show that $T(K) = \mathcal{O}(\frac{\sqrt{K}}{1+\log K})$ via integral inequality and integration by parts:

$$T(K) = \sum_{j=1}^{K} \frac{1}{\sqrt{j}(1 + \log j)} = \mathcal{O}(1) + \sum_{j=\lceil e^2 \rceil}^{K} \frac{1}{\sqrt{j}(1 + \log j)}$$

$$\leq \mathcal{O}(1) + \int_{e^2}^{K} \frac{dz}{\sqrt{z}(1 + \log z)} = \mathcal{O}(1) + 2 \int_{e}^{\sqrt{K}} \frac{du}{1 + 2\log u}$$

where the last step uses a variable substitution $z = u^2$ ($dz = 2u\,du$). Continuing from the integral:

$$\int_{e}^{\sqrt{K}} \frac{du}{1 + 2\log u} = \left[ \frac{u}{1 + 2\log u} \right]_{u=e}^{u=\sqrt{K}} + \int_{e}^{\sqrt{K}} \frac{2}{(1 + 2\log u)^2}\,du$$

$$\leq \frac{\sqrt{K}}{1 + \log K} + 2 \int_{e}^{\sqrt{K}} \frac{du}{(1 + 2\log u)^2}$$

$$\leq \frac{\sqrt{K}}{1 + \log K} + \frac{2}{3} \int_{e}^{\sqrt{K}} \frac{du}{1 + 2\log u} \,,$$

where the last step follows from $f(u) = \frac{1}{1+2\log u}$ taking on values in $(0, 1/3]$ for $u \in [e, \sqrt{K}]$, and so $f(u)^2 \leq f(u)/3$. Solving for the integral, we have that

$$\int_e^{\sqrt{K}} \frac{du}{1+2\log u} \leq \frac{3\sqrt{K}}{1+\log K}$$

and so $T(K) = \mathcal{O}(\frac{\sqrt{K}}{1+\log K})$. Going back to bounding $F$, we have that

$$F \leq 2\sqrt{2}C^2 \sum_{k=1}^{n-1} \frac{T(\lceil k/2 \rceil)}{k^{3/2}(1+\log k)} \leq C'' \sum_{k=1}^{n-1} \frac{1}{k(1+\log k)^2}$$

$$\leq C'' \left(1 + \int_1^\infty \frac{dz}{z(1+\log z)^2}\right) = C'' \left(1 + \int_0^\infty \frac{du}{(1+u)^2}\right) = 2C'',$$

where the integral step uses the variable substitution $u = \log z$ ($dz = z\,du$), and $C''$ is an absolute constant.

**Bounding $N$.**  Here we have that $\lceil k/2 \rceil + 1 \leq j < k$ (and so $k - j \leq k/2 - 1$). It follows that

$$H(j, k) = \sum_{r=1}^j \frac{1}{\sqrt{r(r+k-j)}} \leq \frac{1}{\sqrt{1+k-j}} + \int_1^j \frac{dz}{\sqrt{z(z+k-j)}}$$

$$= \frac{1}{\sqrt{1+k-j}} + \left[2\operatorname{arsinh}\sqrt{\frac{z}{k-j}}\right]_{z=1}^{z=j} \leq \frac{1}{\sqrt{1+k-j}} + 2\operatorname{arsinh}\sqrt{\frac{j}{k-j}}.$$

Noting that $\operatorname{arsinh}(u) \leq \log(1+2u) \leq \log 3u$ for $u \geq 1$, and that $\frac{k}{k-j} \geq \frac{j}{k-j} \geq 1$, we can simplify further:

$$H(j, k) \leq \frac{1}{\sqrt{1+k-j}} + 2\log\left(3\sqrt{\frac{j}{k-j}}\right) \leq 2\log\left(\frac{9k}{k-j}\right)$$

Plugging our bound into the expression for $N$ yields:

$$N = \sum_{k=1}^{n-1} \sum_{j=\lceil k/2 \rceil+1}^{k-1} \Delta_j \Delta_k H(j, k)$$

$$\leq \sum_{k=1}^{n-1} \sum_{d=1}^{\lceil k/2 \rceil} \Delta_k \Delta_{k-d} H(k, k-d)$$

$$\leq 2C^2 \sum_{k=1}^{n-1} \sum_{d=1}^{\lceil k/2 \rceil} \frac{\log(9k/d)}{k(1+\log k)(k-d)(1+\log(k-d))}$$

$$\leq 4C^2 \sum_{k=1}^{n-1} \frac{1}{k^2(1+\log(k/2))^2} \sum_{d=1}^{\lceil k/2 \rceil} \log(9k/d).$$

For the inner sum, we note that

$$\sum_{d=1}^{\lceil k/2 \rceil} \log(9k/d) = \lceil k/2 \rceil \log(9k) - \log(\lceil k/2 \rceil!)$$

$$\leq \lceil k/2 \rceil \log(9k) - \lceil k/2 \rceil \log(\lceil k/2 \rceil) + \lceil k/2 \rceil$$

$$\leq (1 + \log 18)\lceil k/2 \rceil$$

where the first inequality uses Stirling's lower bound: $\log(t!) \geq t\log t - t$. Continuing, we have thus shown

$$N \leq 4C^2(1+\log 18) \sum_{k=1}^{n-1} \frac{\lceil k/2 \rceil}{k^2(1+\log(k/2))^2} \leq C''' \sum_{k=1}^{n-1} \frac{1}{k(1+\log k)^2}$$

$$\leq C''' \left(1 + \int_1^\infty \frac{1}{k(1+\log k)^2}\right) \leq 2C'''$$

where the last integral was already computed in bounding $F$, and $C'''$ is an absolute constant. Taking it all together, we have shown that

$$S = D + 2F + 2N = \mathcal{O}(1),$$

proving the theorem statement. $\qquad\square$

**Theorem 1.** *Let $(\chi_t)_{t=1}^n$ be a sequence on $[\beta, 1]$ for some constant $\beta > 0$. For $n \geq 2$ we define*

$$\Delta_t = |\chi_t - \chi_{t+1}| \qquad (\text{for all } 1 \leq t \leq n-1). \tag{8}$$

*If either of the following two conditions holds ($c > 0$ an absolute constant):*

$$\Delta_t \leq \frac{c}{t(1 + \log t)} \qquad (\text{for all } 1 \leq t \leq n-1), \qquad \text{or} \qquad \sum_{t=1}^{n-1} \Delta_t^2 = o\left(\frac{\log n}{n}\right), \tag{9}$$

*then the factorization $B_\chi \times A_1^{1/2}$, where $B_\chi := A_\chi (A_1)^{-1/2}$, satisfies*

$$\mathrm{MaxSE}(B_\chi, A_1^{1/2}) = \Theta\left(\sqrt{\log n} \cdot \sqrt{\max_{m \in [n]} \chi_m^2 \log m}\right), \tag{10}$$

$$\mathrm{MeanSE}(B_\chi, A_1^{1/2}) = \Theta\left(\sqrt{\log n} \cdot \sqrt{\frac{1}{n} \sum_{m=1}^n \chi_m^2 \log m}\right). \tag{11}$$

*Proof.* Statement follows immediately from invoking Lemma 7 with the bounds on $Q$ derived from Lemma 8 and 9. $\qquad\square$

**Lemma 1.** *Every learning rate schedule $(\chi_t)_{t=1}^n$ with constant $\beta \in (0, 1/e)$ presented in Table 1 satisfies the assumptions of Theorem 1.*

*Proof.* The result will be derived from invoking Corollary 1. We split the treatment of the learning schedules based on if their change over time is, roughly, uniform (exponential/linear/cosine schedules), or not (polynomial schedule). For the first case we show that $\|\Delta\|_2 = \sqrt{\sum_{t=1}^{n-1} \Delta_t^2} = o\left(\sqrt{\log(n)/n}\right)$; for the second we show that $\Delta_t = \mathcal{O}(1/(t \log t))$. We begin with the uniform case.

**Exponential schedule.** $\chi_k = \beta^{\frac{k-1}{n-1}}$ and so

$$\Delta_t = |\chi_t - \chi_{t+1}| = \beta^{\frac{t-1}{n-1}}(1 - \beta^{\frac{1}{n-1}}) \leq 1 - e^{-\frac{\log(1/\beta)}{n-1}} = \mathcal{O}\left(\frac{\log(1/\beta)}{n}\right).$$

It follows that $\|\Delta\|_2 = \mathcal{O}(\log(1/\beta)/\sqrt{n}) = o\left(\sqrt{\log(n)/n}\right)$,

**Linear schedule.** $\chi_k = 1 - (1-\beta)\frac{k-1}{n-1}$ and so

$$\Delta_t = |\chi_t - \chi_{t+1}| = \frac{1-\beta}{n-1}$$

and so $\|\Delta\|_2 = \mathcal{O}(1/\sqrt{n-1}) = o\left(\sqrt{\log(n)/n}\right)$.

**Cosine schedule.** $\chi_k = \beta + \frac{1-\beta}{2}\left(1 + \cos\left(\frac{(k-1)}{n-1}P\pi\right)\right)$, and so

$$\Delta_t = |\chi_t - \chi_{t+1}| = \frac{1-\beta}{2}\left|\cos\left(\frac{t-1}{n-1}\pi\right) - \cos\left(\frac{t}{n-1}\pi\right)\right|$$

$$= \frac{(1-\beta)}{2} \cdot \frac{\pi}{n-1}\left|\sin\left(\frac{\xi}{n-1}\pi\right)\right| \leq \frac{(1-\beta)\pi}{2(n-1)} = \mathcal{O}(1/n)$$

where the third equality uses the mean value theorem applied to $f(z) = \cos(cz)$ on $[t-1, t]$ with $\xi \in (t-1, t)$. It follows that $\|\Delta\|_2 = \mathcal{O}(1/\sqrt{n})$, which is $o\left(\sqrt{\log(n)/n}\right)$.

Table 5: MaxSE and MeanSE errors for the factorization $B_\chi \times A_1^{1/2} = A_\chi(A_1)^{-1/2} \times A_1^{1/2} = A_\chi$ under single participation, listed for each of the learning rate schedules in Table 1. $\beta \in (0, 1/e)$ is assumed throughout. (a) Exponential decay, proven in Corollary 6; (b) polynomial decay, proven in Corollary 7; (c) linear decay, proven in Corollary 8; (d) cosine decay, proven in Corollary 9.

| Learning rate $\chi_t$ | MaxSE | MeanSE |
|---|---|---|
| (a) $\chi_t = \beta^{\frac{t-1}{n-1}}$ | $\Theta\left(\sqrt{\log n}\sqrt{\log \frac{n}{\log(1/\beta)}}\right)$ | $\Theta\left(\frac{\log n}{\sqrt{\log(1/\beta)}}\right)$ |
| (b) $\chi_t = \beta + (1-\beta)\frac{(n/t)^\gamma - 1}{n^\gamma - 1}, \gamma \geq 1$ | $\Theta\left(\sqrt{\log n\left(\beta^2 \log n + \frac{(1-\beta)^2}{\gamma}\right)}\right)$ | $\Theta(\beta \log n)$ |
| (c) $\chi_t = \beta + (1-\beta)\frac{t-1}{n-1}$ | $\Theta(\log n)$ | $\Theta(\log n)$ |
| (d) $\chi_t = \beta + \frac{1-\beta}{2}\left(1 + \cos\left(\frac{t-1}{n-1}\pi\right)\right)$ | $\Theta(\log n)$ | $\Theta(\log n)$ |

**Polynomial schedule.** $\chi_k = \beta + (1-\beta)\frac{\left(\frac{n}{k}\right)^\gamma - 1}{n^\gamma - 1}, \gamma > 0$, and so

$$\Delta_k = |\chi_k - \chi_{k+1}| = \frac{1-\beta}{n^\gamma - 1}\left[\left(\frac{n}{k}\right)^\gamma - \left(\frac{n}{k+1}\right)^\gamma\right]$$

$$= \frac{(1-\beta)n^\gamma}{(n^\gamma - 1)k^\gamma}\left[1 - \left(\frac{k}{k+1}\right)^\gamma\right] = \mathcal{O}\left(k^{-(\gamma+1)}\right).$$

This also implies $\Delta_k = \mathcal{O}\left(\frac{1}{k \log k}\right)$, completing the last case. $\qquad\square$

### E.1 ERROR FOR SPECIFIC LEARNING RATES

In this section we give tight error bounds for the prefix-sum factorization for each of the learning rate schedules discussed in this paper (see Table 1 for the list). In Table 5 we give the corresponding error bounds, all of which are proved later in the section.

#### E.1.1 EXPONENTIAL LEARNING RATE DECAY

**Lemma 10.** *Let $\beta \in (0, 1/e)$ and $\alpha = \beta^{1/(n-1)}$. Then*

$$\max_{1 \leq m \leq n} \alpha^m \log m = \Theta\left(\log \frac{n}{\log(1/\beta)}\right). \tag{59}$$

*Proof.* For the lower bound, take $m_0 = \lceil 1/\log(1/\alpha)\rceil$. Since $\log(1/\alpha) = \frac{1}{n-1}\log(1/\beta)$, we have $m_0 \leq (n-1)/\log(1/\beta) < n$, so $m_0$ is admissible. Moreover, $\alpha^{m_0} \geq e^{-1}\alpha$ and $\log m_0 \geq \log\frac{1}{\log(1/\alpha)}$, giving

$$\max_{1 \leq m \leq n} \alpha^m \log m \geq \Omega\left(\log \frac{1}{\log(1/\alpha)}\right).$$

For the upper bound, write $f(m) = \alpha^m \log m$ with real $m > 1$. Then $\frac{d}{dm}\log f(m) = \log \alpha + 1/(m \log m)$, so the maximizer satisfies $m \log m = 1/\log(1/\alpha)$. At this point, $\log m \sim \log\frac{1}{\log(1/\alpha)}$ and $\alpha^m = e^{-1/\log m} = \Theta(1)$, hence $f(m) = \mathcal{O}(\log\frac{1}{\log(1/\alpha)})$.

Thus

$$\max_{1 \leq m \leq n} \alpha^m \log m = \Theta\left(\log \frac{1}{\log(1/\alpha)}\right).$$

Finally, since $\log\frac{1}{\log(1/\alpha)} = \log\frac{n-1}{\log(1/\beta)} = \Theta(\log\frac{n}{\log(1/\beta)})$, the claim follows. $\qquad\square$

**Lemma 11.** *Let $\beta \in (0, 1/e)$ and $\alpha = \beta^{1/(n-1)}$. Then*

$$\frac{1}{n}\sum_{m=1}^{n} \alpha^m \log m = \Theta\left(\frac{\log n}{\log(1/\beta)}\right). \tag{60}$$

*Proof.* Splitting $\log m = \log n + \log(m/n)$ gives

$$\frac{1}{n} \sum_{m=1}^{n} \alpha^m \log m = \frac{\log n}{n} \sum_{m=1}^{n} \alpha^m + \frac{1}{n} \sum_{m=1}^{n} \alpha^m \log(m/n).$$

The first sum is geometric: $\sum_{m=1}^{n} \alpha^m = \alpha(1 - \alpha^n)/(1 - \alpha)$. Since $\alpha = 1 - \frac{\log(1/\beta)}{n-1} + o(1/n)$, we have $1 - \alpha \sim \frac{\log(1/\beta)}{n-1}$ and $\alpha^n \to \beta$. Thus $\frac{1}{n} \sum_{m=1}^{n} \alpha^m \sim (1 - \beta)/\log(1/\beta)$, so the first term is $\sim \frac{1-\beta}{\log(1/\beta)} \log n = \Theta(\frac{\log n}{\log(1/\beta)})$.

The second sum is a Riemann sum, converging to $I(\beta) = \int_0^1 \beta^x \log x \, dx$. Since $I$ is monotone decreasing with $I(0) = 0$, $I(1) = -1$, we have $I(\beta) = \Theta(1)$. Hence the first term dominates, and the result follows. $\qquad \square$

**Corollary 6.** *For exponential learning rate decay $\chi_k = \beta^{\frac{k-1}{n-1}}$ with $\beta \in (0, 1/e)$, the prefix-sum–based factorization $A_\chi = A_\chi (A_1)^{-1/2} \times A_1^{1/2}$ gives the following values for MaxSE and MeanSE:*

$$\text{MaxSE}(B_\chi, A_1^{1/2}) = \Theta\left( \sqrt{\log n} \sqrt{\log \frac{n}{\log(1/\beta)}} \right), \tag{61}$$

$$\text{MeanSE}(B_\chi, A_1^{1/2}) = \Theta\left( \frac{\log n}{\sqrt{\log(1/\beta)}} \right). \tag{62}$$

*Proof.* Invoking Lemma 1 we have that

$$\text{MaxSE}\left(B_\chi, A_1^{1/2}\right) = \Theta\left( \sqrt{\log n} \cdot \sqrt{\max_{1 \leq t \leq n} \beta^{2\frac{t-1}{n-1}} \log t} \right) = \Theta\left( \sqrt{\log n} \sqrt{\frac{n}{\log(1/\beta)}} \right)$$

$$\text{MeanSE}\left(B_\chi, A_1^{1/2}\right), = \Theta\left( \sqrt{\log n} \cdot \sqrt{\frac{1}{n} \sum_{t=1}^{n} \beta^{\frac{2(t-1)}{n-1}} \log t} \right) = \Theta\left( \sqrt{\frac{\log n}{\log(1/\beta)}} \right),$$

where the last step of each equation invokes Lemma 10 and 11 respectively for $\beta' = \beta^2 \in (0, 1/e)$. $\qquad \square$

### E.1.2 POLYNOMIAL LEARNING RATE DECAY

**Lemma 12.** *Let $1 \leq m \leq n$ be integers, $\beta \in (0, 1/e)$, $\gamma \geq 1$ and $n$ sufficiently large. Let $\chi_k = \beta + (1 - \beta)\frac{\left(\frac{n}{k}\right)^\gamma - 1}{n^\gamma - 1}$. Then*

$$\max_{1 \leq m \leq n} \chi_m^2 \log m = \Theta\left( \beta^2 \log n + \frac{(1 - \beta)^2}{\gamma} \right).$$

*Proof.* Before we start, we will find the following inequality useful:

$$\chi_m = \beta + (1 - \beta)m^{-\gamma} + \frac{(1 - \beta)(m^{-\gamma} - 1)}{n^\gamma - 1} \geq \left( 1 - \frac{1 - \beta}{\beta(n^\gamma - 1)} \right) \left( \beta + (1 - \beta)m^{-\gamma} \right)$$

In particular for large enough $n$, and $1 \leq m \leq n$, we have that

$$\chi_m \geq \frac{1}{2} \left( \beta + (1 - \beta)m^{-\gamma} \right),$$

and for all $n$, and $1 \leq m \leq n$, also that

$$\chi_m \leq \beta + (1 - \beta)m^{-\gamma},$$

implying that it suffices for us to argue about

$$\chi_m = \Theta(\beta + (1 - \beta)m^{-\gamma}) \tag{63}$$

when convenient. We now begin with the upper bound. Using $(a+b)^2 \leq 2a^2 + 2b^2$ in the first step:

$$\max_{1 \leq m \leq n} \chi_m^2 \log m \leq \max_{1 \leq m \leq n} 2(\beta^2 + (1-\beta)^2 m^{-2\gamma}) \log m$$

$$\leq 2\beta^2 \log n + 2(1-\beta)^2 \max_{1 \leq m \leq n} m^{-2\gamma} \log m$$

Defining $f(z) = z^{-2\gamma} \log z$, we have that $f'(z) = z^{-2\gamma-1}(1 - 2\gamma \log z)$. Solving $f(z) = 0$ yields the maximizer $z = e^{\frac{1}{2\gamma}}$, and so

$$\max_m \chi_m^2 \log m \leq 2 \left( \beta^2 \log n + \frac{(1-\beta)^2}{2e\gamma} \right) = \mathcal{O}\left( \beta^2 \log n + \frac{(1-\beta)^2}{\gamma} \right).$$

For the lower bound, we note that setting $m_0 = n$ yields $\chi_{m_0}^2 \log m_0 = \beta^2 \log n$.

Instead choosing $m_0 = \left\lceil e^{\frac{1}{2\gamma}} \right\rceil$ yields

$$\chi_{m_0}^2 \log m_0 \geq \frac{1}{4}(\beta + (1-\beta)e^{-\frac{1}{2}})^2 \log(e^{\frac{1}{2\gamma}} - 1) \geq \frac{(1-\beta)^2}{4e} \log(e^{\frac{1}{2\gamma}} - 1) = \Omega\left( \frac{(1-\beta)^2}{\gamma} \right)$$

Combining the two lower bounds, we get

$$\max_m \chi_m^2 \log m = \Omega\left( \max\left\{ \beta^2 \log n, \frac{(1-\beta)^2}{\gamma} \right\} \right) = \Omega\left( \beta^2 \log n + \frac{(1-\beta)^2}{\gamma} \right),$$

finishing the proof. $\qquad\square$

**Lemma 13.** *Let $1 \leq m \leq n$ be integers, $\beta \in (0, 1/e)$, $\gamma \geq 1$ and $n$ sufficiently large. Let $\chi_k = \beta + (1-\beta)\frac{\left(\frac{n}{k}\right)^\gamma - 1}{n^\gamma - 1}$. Then*

$$\frac{1}{n} \sum_{m=1}^{n} \chi_m^2 \log m = \Theta(\beta^2 \log n).$$

*Proof.* We will again use that

$$\frac{1}{2}(\beta + (1-\beta)^2 m^{-\gamma}) \leq \chi_m \leq \beta + (1-\beta)^2 m^{-\gamma}$$

as shown in the proof of Lemma 12. First the upper bound. We write

$$\frac{1}{n} \sum_{m=1}^{n} \chi_m^2 \log m \leq \frac{1}{n} \sum_{m=1}^{n} 2(\beta^2 + (1-\beta)^2 m^{-2\gamma}) \log m \leq \frac{2\log n}{n} \sum_{m=2}^{n} \beta^2 + (1-\beta)^2 m^{-2\gamma}$$

$$\leq 2\beta^2 \log n + \frac{2(1-\beta)^2 \log n}{n} \int_{t=1}^{n} t^{-2\gamma} \, dt$$

$$= 2\beta^2 \log n + \frac{2(1-\beta)^2 \log n}{n} \cdot \frac{n^{2\gamma-1} - 1}{2\gamma - 1} = \mathcal{O}(\beta^2 \log n),$$

where the second term in the second-to-last expression can be identified as $o(\log n)$ for any value of $\gamma > 0$. For the lower bound,

$$\frac{1}{n} \sum_{m=1}^{n} \chi_m^2 \log m \geq \frac{1}{n} \sum_{m=1}^{n} \frac{1}{4} \left( \beta + (1-\beta)m^{-\gamma} \right)^2 \log m \geq \frac{1}{4n} \sum_{m=1}^{n} \beta^2 \log m$$

$$\geq \frac{\beta^2}{4n} \sum_{m=\lceil m/2 \rceil}^{n} \log m \geq \frac{\beta^2 (n/2 - 1) \log(n/2)}{4n} = \Omega(\beta^2 \log n). \qquad\square$$

**Corollary 7.** *For polynomial learning rate decay $\chi_k = \beta + (1-\beta)\frac{\left(\frac{n}{k}\right) - 1}{n^\gamma - 1}$ with constant $\beta \in (0, 1/e)$ and $\gamma \geq 1$, the prefix-sum-based factorization $A_\chi = A_\chi(A_1^{-1/2}) \times A_1^{1/2}$ gives the following values for MaxSE and MeanSE:*

$$\mathrm{MaxSE}(B_\chi, A_1^{-1/2}) = \Theta\left( \sqrt{\log n \left( \beta^2 \log n + \frac{(1-\beta)^2}{\gamma} \right)} \right),$$

$$\mathrm{MeanSE}(B_\chi, A_1^{-1/2}) = \Theta\left( \beta \log n \right).$$

*Proof.* Result is immediate from invoking Lemma 1, together with Lemma 12 and 13 for the MaxSE and MeanSE errors respectively. □

### E.1.3 LINEAR LEARNING RATE DECAY

**Lemma 14.** *Let $\chi_k = 1 - (1 - \beta)\frac{k-1}{n-1}$, $\beta \in (0, 1/e)$ and $n \geq 2$. Then*

$$\max_{1 \leq m \leq n} \chi_m^2 \log m = \Theta(\log n).$$

*Proof.* For the upper bound, using $\chi_k \leq 1$, we directly get

$$\max_{1 \leq m \leq n} \chi_m^2 \log m \leq \log n.$$

For the lower bound, pick $m_0 = \lfloor (n+1)/2 \rfloor$ where $\chi_{m_0} \geq (1 + \beta)/2$:

$$\max_{1 \leq m \leq n} \chi_m^2 \log m \geq \chi_{m_0}^2 \log m_0 = \frac{(1+\beta)^2}{4} \log \left\lfloor \frac{n+1}{2} \right\rfloor = \Omega(\log n),$$

finishing the proof. □

**Lemma 15.** *Let $\chi_k = 1 - (1 - \beta)\frac{k-1}{n-1}$, $\beta \in (0, 1/e)$ and $n \geq 2$. Then*

$$\frac{1}{n} \sum_{m=1}^{n} \chi_m^2 \log m = \Theta(\log n).$$

*Proof.* For the upper bound, again using $\chi_k \leq 1$, and $\log m \leq \log n$ we directly get

$$\frac{1}{n} \sum_{m=1}^{n} \chi_m^2 \log m \leq \frac{1}{n} \cdot n \log n = \log n.$$

For the lower bound, we truncate the sum at $m_0 = \lfloor (n+1)/2 \rfloor$ and use the bound $\chi_k \geq \frac{1+\beta}{2}$ for all $k \leq m_0$:

$$\frac{1}{n} \sum_{m=1}^{n} \chi_m^2 \log m \geq \frac{1}{n} \sum_{m=1}^{m_0} \chi_m^2 \log m \geq \frac{(1+\beta)^2}{4n} \sum_{m=1}^{m_0} \log m \geq \Omega(\log n),$$

where the last step can be seen by truncating the sum, taking the upper half of the indices, and lower bounding each of the $\Omega(n)$ logarithms by $\log \lceil m_0/2 \rceil = \Omega(\log n)$. This finishes the proof. □

**Corollary 8.** *For linear learning rate decay $\chi_k = 1 - (1 - \beta)\frac{k-1}{n-1}$ with $\beta \in (0, 1/e)$, the prefix-sum–based factorization $A_\chi = A_\chi(A_1)^{-1/2} \times A_1^{1/2}$ gives the following values for MaxSE and MeanSE:*

$$\text{MaxSE}(B_\chi, A_1^{1/2}) = \Theta(\log n), \qquad \text{MeanSE}(B_\chi, A_1^{1/2}) = \Theta(\log n). \tag{64}$$

*Proof.* Result is immediate from invoking Lemma 1, together with Lemma 14 and 15 for the MaxSE and MeanSE errors respectively. □

### E.1.4 COSINE LEARNING RATE DECAY.

**Lemma 16.** *Let $1 \leq m \leq n$ be integers, $\beta \in (0, 1/e)$, and $n$ sufficiently large. Let $\chi_k = \beta + \frac{1-\beta}{2}(1 + \cos(\frac{k-1}{n-1}\pi))$. Then*

$$\max_{1 \leq m \leq n} \chi_m^2 \log m = \Theta(\log n).$$

*Proof.* First the upper bound. We use that $\chi_k \leq 1$:

$$\max_{1 \leq m \leq n} \chi_m^2 \log m \leq \max_{1 \leq m \leq n} \log m = \log n\,.$$

For the lower bound, we set $m_0 = \lfloor (n+1)/2 \rfloor$.

$$\max_{1 \leq m \leq n} \chi_m^2 \log m \geq \left( \beta + \frac{1-\beta}{2} \left( 1 + \cos \left( \frac{\lfloor (n+1)/2 \rfloor - 1}{n-1} \pi \right) \right) \right)^2 \log \left\lfloor \frac{n+1}{2} \right\rfloor$$

$$\geq \left( \beta + \frac{1-\beta}{2} \left( 1 + \cos \left( \frac{\pi}{2} \right) \right) \right)^2 \log \left( \frac{n-1}{2} \right)$$

$$= \frac{(1+\beta)^2}{4} \log \left( \frac{n-1}{2} \right) = \Omega(\log n)\,. \qquad \square$$

**Lemma 17.** *Let $1 \leq m \leq n$ be integers, $\beta \in (0, 1/e)$, and $n$ sufficiently large. Let $\chi_k = \beta + \frac{1-\beta}{2} \left( 1 + \cos \left( \frac{k-1}{n-1} \pi \right) \right)$. Then*

$$\frac{1}{n} \sum_{m=1}^{n} \chi_m^2 \log m = \Theta(\log n)\,.$$

*Proof.* For the upper bound, we again use that $\chi_k \leq 1$.

$$\frac{1}{n} \sum_{m=1}^{n} \chi_m^2 \log m \leq \frac{1}{n} \cdot n \log n = \log n$$

We prove the lower bound by truncating the sum.

$$\frac{1}{n} \sum_{m=1}^{n} \chi_m^2 \log m \geq \frac{1}{n} \sum_{m=\lceil n/4 \rceil}^{\lfloor (n+1)/2 \rfloor} \left( \beta + \frac{(1-\beta)}{2} \left( 1 + \cos \left( \frac{m-1}{n-1} \pi \right) \right) \right)^2 \log m$$

$$\geq \frac{1}{n} \sum_{m=\lceil n/4 \rceil}^{\lfloor (n+1)/2 \rfloor} \left( \beta + \frac{(1-\beta)}{2} \left( 1 + \cos \left( \frac{\lfloor (n+1)/2 \rfloor - 1}{n-1} \pi \right) \right) \right)^2 \log \left\lceil \frac{n}{4} \right\rceil$$

$$\geq \frac{1}{n} \sum_{m=\lceil n/4 \rceil}^{\lfloor (n+1)/2 \rfloor} \left( \frac{1+\beta}{2} \right)^2 \log \left( \frac{n}{4} \right)$$

$$= \frac{(1+\beta)^2}{4n} \left( \left\lfloor \frac{n+1}{2} \right\rfloor - \left\lceil \frac{n}{4} \right\rceil + 1 \right) \log \left( \frac{n}{4} \right) = \Omega(\log n). \qquad \square$$

**Corollary 9.** *For cosine learning rate decay $\chi_k = \beta + \frac{1-\beta}{2} \left( 1 + \cos \left( \frac{k-1}{n-1} \pi \right) \right)$ with $\beta \in (0, 1/e)$, the prefix-sum-based factorization $A_\chi = A_\chi (A_1)^{-1/2} \times A_1^{1/2}$ gives the following values for MaxSE and MeanSE:*

$$\mathrm{MaxSE}(B_\chi, A_1^{-1/2}) = \Theta(\log n)\,, \qquad \mathrm{MeanSE}(B_\chi, A_1^{-1/2}) = \Theta(\log n)\,.$$

*Proof.* Result is immediate from invoking Lemma 1, together with Lemma 16 and 17 for the MaxSE and MeanSE errors respectively. $\qquad \square$

## F  SINGLE-PARTICIPATION: LOWER BOUNDS

**Theorem 2.** *Let $A_\chi = A_1 D_\chi$, where $D_\chi = \mathrm{diag}(\chi_1, \ldots, \chi_n)$ with positive $\chi_t > 0$. Then*

$$\inf_{B \times C = A_\chi} \textit{MaxSE}(B, C) \geq \max_{1 \leq t \leq n} \frac{1}{\pi} \left( \min_{j \leq t} \chi_j \right) \log t \tag{12}$$

$$\inf_{B \times C = A_\chi} \textit{MeanSE}(B, C) \geq \max_{1 \leq t \leq n} \frac{1}{\pi} \sqrt{\frac{t}{n}} \left( \min_{j \leq t} \chi_j \right) \log t\,. \tag{13}$$

*Proof.* We prove each bound separately in Lemmas 18 and 19. $\qquad\square$

**Lemma 18.** *Let $A_\chi = A_1 D_\chi$, where $D_\chi = \mathrm{diag}(\chi_1, \ldots, \chi_n)$ with positive $\chi_1, \chi_2, \ldots, \chi_n$. Then*

$$\gamma_2(A_\chi) := \inf_{B \times C = A_\chi} MaxSE(B, C) \ \geq\ \max_{1 \leq k \leq n} \frac{1}{\pi} \left(\min_{j \leq k} \chi_j\right) \log k.$$

*Proof.* The optimal factorization error can be written as

$$\gamma_2(A) = \max\left\{ \|P^{1/2} A Q^{1/2}\|_* : P, Q \text{ diag., nonneg., } \mathrm{Tr}\, P = \mathrm{Tr}\, Q = 1 \right\},$$

where $\|\cdot\|_*$ denotes the nuclear norm of a matrix. It was observed in Matoušek et al. (2020) that this norm is monotonic with respect to taking submatrices: if $A = B \times C$, then removing rows from $B$ cannot increase the maximum row norm, and removing columns from $C$ cannot increase the maximum column norm. Thus, we can lower bound $\gamma_2(A)$ by the $k \times k$ principal submatrix consisting of the first $k$ rows and columns:

$$\gamma_2(A_\chi) \ \geq\ \gamma_2((A_\chi)_{:k,:k}).$$

For the lower bound, let us assume $P = \frac{1}{k} I_k$ and $Q = \frac{1}{\mathrm{Tr}(D_{:k,:k}^{-2})} D_{:k,:k}^{-2}$, which gives

$$\gamma_2((A_\chi)_{:k,:k}) \ \geq\ \frac{\|(A_1)_{:k,:k}\|_*}{\sqrt{k} \sqrt{\sum_{j=1}^{k} \chi_j^{-2}}}.$$

Using the bound $\|(A_1)_{:k,:k}\|_* \geq \frac{k}{\pi} \log k$ and the fact that $\sum_{j=1}^{k} \chi_j^{-2} \leq k (\min_{j \leq k} \chi_j)^{-2}$, we conclude

$$\gamma_2(A_\chi) \ \geq\ \frac{1}{\pi} \left(\min_{j \leq k} \chi_j\right) \log k.$$

Maximizing over $k$ yields the lemma. $\qquad\square$

**Lemma 19.** *Let $A_\chi = A_1 D_\chi$, where $D_\chi = \mathrm{diag}(\chi_1, \ldots, \chi_n)$ with positive $\chi_1, \chi_2, \ldots, \chi_n$. Then*

$$\gamma_F(A_\chi) \ =\ \inf_{A_\chi = BC} \mathrm{MeanSE}(B, C) \ \geq\ \max_{1 \leq k \leq n} \frac{1}{\pi} \sqrt{\frac{k}{n}} \left(\min_{j \leq k} \chi_j\right) \log k.$$

*Proof.* By definition,

$$\gamma_F(A) = \inf_{A = BC} \frac{1}{\sqrt{n}} \|B\|_F \|C\|_{1 \to 2},$$

where $\|C\|_{1 \to 2} = \max_j \|C_{:,j}\|_2$ is the maximum column norm.

Fix $k \leq n$. For any factorization $A = BC$, the principal $k \times k$ submatrix satisfies

$$A_{:k,:k} \ =\ B_{:k,:} C_{:k}.$$

Since removing rows can only decrease the Frobenius norm, $\|B_{:k,:}\|_F \leq \|B\|_F$, and removing columns can only decrease the $\|\cdot\|_{1 \to 2}$ norm, $\|C_{:k}\|_{1 \to 2} \leq \|C\|_{1 \to 2}$. Therefore

$$\frac{1}{\sqrt{n}} \|B\|_F \|C\|_{1 \to 2} \ \geq\ \frac{1}{\sqrt{n}} \|B_{:k,:}\|_F \|C_{:k}\|_{1 \to 2} \ =\ \sqrt{\frac{k}{n}} \left( \frac{1}{\sqrt{k}} \|B_{:k,:}\|_F \|C_{:k}\|_{1 \to 2} \right).$$

Taking the infimum over all factorizations gives

$$\gamma_F(A) \ \geq\ \sqrt{\frac{k}{n}} \gamma_F(A_{:k,:k}). \tag{65}$$

For the submatrix, we use the bound from Henzinger et al. (2023):

$$\gamma_F((A_\chi)_{:k,:k}) \ \geq\ \frac{\|(A_\chi)_{:k,:k}\|_*}{k}, \tag{66}$$

where $\| \cdot \|_*$ denotes the nuclear norm. Recall that the nuclear norm is dual to the spectral norm:

$$\|M\|_* \;=\; \sup_{\|Y\|_2 \leq 1} \mathrm{tr}(MY^\top),$$

where the supremum is over all matrices $Y$ with operator norm at most 1. Write $(A_\chi)_{:k,:k} = (A_1)_{:k,:k} D_k$ with $D_k = \mathrm{diag}(\chi_1, \ldots, \chi_k)$. If $W$ is a dual certificate for $(A_1)_{:k,:k}$, so that $\|W\|_2 \leq 1$ and $\|(A_1)_{:k,:k}\|_* = \mathrm{tr}((A_1)_{:k,:k} W^\top)$, then consider

$$Y = \frac{W D_k^{-1}}{\|D_k^{-1}\|_2}.$$

Since $\|W\|_2 \leq 1$, we have $\|Y\|_2 \leq 1$. Thus

$$\|(A_\chi)_{:k,:k}\|_* \;\geq\; \mathrm{tr}((A_1)_{:k,:k} D_k Y^\top) = \frac{1}{\|D_k^{-1}\|_2} \mathrm{tr}((A_1)_{:k,:k} W^\top) = \frac{1}{\|D_k^{-1}\|_2} \|(A_1)_{:k,:k}\|_*.$$

The largest diagonal entry of $D_k^{-1}$ is $(\min_{j \leq k} \chi_j)^{-1}$, so

$$\|(A_\chi)_{:k,:k}\|_* \;\geq\; (\min_{j \leq k} \chi_j) \|(A_1)_{:k,:k}\|_*. \tag{67}$$

Finally, using the standard estimate $\|(A_1)_{:k,:k}\|_* \geq \frac{k}{\pi} \log k$, combining (65), (66), and (67) gives

$$\gamma_F(A_\chi) \;\geq\; \sqrt{\frac{k}{n}} \cdot \frac{1}{k} \cdot \frac{k}{\pi} (\min_{j \leq k} \chi_j) \log k \;=\; \frac{1}{\pi} \sqrt{\frac{k}{n}} (\min_{j \leq k} \chi_j) \log k.$$

Maximizing over $k$ proves the lemma. $\qquad\square$

**Corollary 2.** *Suppose* $\chi_k = \beta^{\frac{k-1}{n-1}}$ *with* $\beta \in (0, 1/e)$. *Then*

$$\inf_{B \times C = A_\chi} MaxSE(B, C) = \Omega\left( \log \frac{n}{\log(1/\beta)} \right) \tag{16}$$

$$\inf_{B \times C = A_\chi} MeanSE(B, C) = \Omega\left( \frac{1}{\sqrt{\log(1/\beta)}} \log \frac{n}{\log(1/\beta)} \right). \tag{17}$$

*Proof.* By Theorem 2, for any $t$,

$$\inf_{A_\chi = BC} \mathrm{MaxSE}(B, C) \;\geq\; \frac{\chi_t}{\pi} \log t, \qquad \inf_{A_\chi = BC} \mathrm{MeanSE}(B, C) \;\geq\; \frac{\chi_t}{\pi} \sqrt{\frac{t}{n}} \log t.$$

Choose

$$t^\star = \left\lceil \frac{n}{\log(1/\beta)} \right\rceil,$$

which satisfies $1 \leq t^\star \leq n$ since $\beta \in (0, 1/e)$. Then

$$\chi_{t^\star} = \beta^{\frac{t^\star - 1}{n-1}} = \exp\left( -\frac{\log(1/\beta)}{n-1} (t^\star - 1) \right) = \Theta(1).$$

Hence

$$\inf_{A_\chi = BC} \mathrm{MaxSE}(B, C) \;\geq\; \frac{\chi_{t^\star}}{\pi} \log t^\star = \Omega\left( \log \frac{n}{\log(1/\beta)} \right),$$

and, using $t^\star/n = \Theta\big(1/\log(1/\beta)\big)$,

$$\inf_{A_\chi = BC} \mathrm{MeanSE}(B, C) \;\geq\; \frac{\chi_{t^\star}}{\pi} \sqrt{\frac{t^\star}{n}} \log t^\star = \Omega\left( \frac{1}{\sqrt{\log(1/\beta)}} \log \frac{n}{\log(1/\beta)} \right).$$

$\qquad\square$

## G   MULTI-PARTICIPATION: PREFIX-SUM FACTORIZATION

**Lemma 20.** *Let $(\chi_t)_{t=1}^n$ be a positive sequence taken from $[\beta, \infty)$ where $\beta > 0$ is a constant, and*

$$Q = \sum_{l=1}^{n-1} \left( \sum_{t=0}^{n-l-1} |\chi_{l+t} - \chi_{l+t+1}| r_t \right)^2 = o(\log n) \,.$$

*Then*

$$\|B_\chi^p\|_{2\to\infty} = \Theta \left( \sqrt{\max_{1\le m\le n} \chi_m^2 \log\left(\min\{m,p\}\right) + \frac{1}{p}\sum_{t=p}^{m-1} \chi_t^2} \right) \,,$$

$$\frac{1}{\sqrt{n}}\|B_\chi^p\|_F = \Theta \left( \sqrt{\frac{1}{n}\sum_{m=1}^n \left[ \chi_m^2 \log\left(\min\{m,p\}\right) + \frac{1}{p}\sum_{t=p}^{m-1} \chi_t^2 \right]} \right) \,.$$

*Proof.* The entries of $B_\chi^p$ can be expressed as follows:

$$(B_\chi^p)_{m,l} = \chi_l + \sum_{t=1}^{\min\{m-l,p-1\}} \tilde{r}_t \chi_{t+l} = \chi_l + \sum_{t=1}^{m-l} \tilde{r}_t \chi_{t+l} \mathbb{1}_{t\le p-1} \,, \tag{68}$$

where again $\tilde{r}_t = \frac{-r_t}{2t-1}$. Following the proof of Lemma 7 and applying summation of parts:

$$(B_\chi^p)_{m,l} = \chi_m \sum_{t=0}^{m-l} \tilde{r}_t \mathbb{1}_{t\le p-1} - \sum_{t=0}^{m-l-1} (\chi_{l+t+1} - \chi_{l+t}) \sum_{j=0}^{t} \tilde{r}_j \mathbb{1}_{j\le p-1}$$

$$= \chi_m r_{\min\{m-l,p-1\}} + \sum_{t=0}^{m-l-1} (\chi_{l+t} - \chi_{l+t+1}) r_{\min\{t,p-1\}} \,.$$

For notational convenience, let $\delta_t = \chi_t - \chi_{t+1}$ and $\Delta_t = |\delta_t|$. We have two distinct cases for these sums: $m - l \le p - 1$ and $m - l > p - 1$. Starting with the first case, we get

$$\left(B_\chi^p\right)_{m,l} = \chi_m r_{m-l} + \sum_{t=0}^{m-l-1} \delta_{l+t} r_t \,.$$

as in the case without bandedness $(p = n)$. For the second case, where $m - l > p - 1$, we get

$$\left(B_\chi^p\right)_{m,l} = \chi_m r_{p-1} + \sum_{t=0}^{p-2} \delta_{l+t} r_t + \underbrace{\sum_{t=p-1}^{m-l-1} \delta_{l+t} r_{p-1}}_{=(\chi_{l+p-1} - \chi_m) r_{p-1}} = \chi_{l+p-1} r_{p-1} + \sum_{t=0}^{p-2} \delta_{l+t} r_t \,.$$

Combining the two expressions, we can express the squared row sums:

$$\sum_{l=1}^m \left(B_\chi^p\right)_{m,l}^2 = \sum_{l=1}^{\max\{m-p,0\}} \left(B_\chi^p\right)_{m,l}^2 + \sum_{l=\max\{m-p,0\}+1}^m \left(B_\chi^p\right)_{m,l}^2$$

$$= \underbrace{\sum_{l=1}^{\max\{m-p,0\}} \left( \chi_{l+p-1} r_{p-1} + \sum_{t=0}^{p-2} \delta_{l+t} r_t \right)^2}_{S_1} + \underbrace{\sum_{l=\max\{m-p,0\}+1}^m \left( \chi_m r_{m-l} + \sum_{t=0}^{m-l-1} \delta_{l+t} r_t \right)^2}_{S_2} \,.$$

We will argue that we can characterize $S_1 + S_2$ tightly. Beginning with upper bounds, using $(a + b)^2 \leq 2a^2 + 2b^2$, and letting $q = \max\{m - p, 0\}$:

$$S_1 \leq \sum_{l=1}^{q} \left( \chi_{l+p-1} r_{p-1} + \sum_{t=0}^{p-2} \Delta_{l+t} r_t \right)^2 \leq 2 \underbrace{r_{p-1}^2 \sum_{l=1}^{q} \chi_{l+p-1}^2}_{P_1} + 2 \underbrace{\sum_{l=1}^{q} \left( \sum_{t=0}^{p-2} \Delta_{l+t} r_t \right)^2}_{Q_1},$$

$$S_2 \leq \sum_{l=q+1}^{m} \left( \chi_m r_{m-l} + \sum_{t=0}^{m-l-1} \Delta_{l+t} r_t \right)^2 \leq 2 \underbrace{\chi_m^2 \sum_{l=q+1}^{m} r_{m-l}^2}_{P_2} + 2 \underbrace{\sum_{l=q+1}^{m-1} \left( \sum_{t=0}^{m-l-1} \Delta_{l+t} r_t \right)^2}_{Q_2}.$$

Repeating the exercise to get a lower bound on $S_1 + S_2$ via $(a + b)^2 \geq \frac{1}{2} a^2 - b^2$:

$$S_1 = \sum_{l=1}^{q} \left( \chi_{l+p-1} r_{p-1} + \sum_{t=0}^{p-2} \delta_{l+t} r_t \right)^2 \geq \frac{1}{2} P_1 - \sum_{l=1}^{q} \left( \sum_{t=0}^{p-2} \delta_{l+t} r_t \right)^2 \geq \frac{1}{2} P_1 - Q_1,$$

$$S_2 = \sum_{l=q+1}^{m} \left( \chi_m r_{m-l} + \sum_{t=0}^{m-l-1} \delta_{l+t} r_t \right)^2 \geq \frac{1}{2} P_2 - \sum_{l=q+1}^{m-1} \left( \sum_{t=0}^{m-l-1} \delta_{l+t} r_t \right)^2 \geq \frac{1}{2} P_2 - Q_2,$$

where the last step in each derivation uses that the expression is made smaller when we replace $\delta_{l+t}$ by $\Delta_{l+t}$. It follows that

$$\sum_{l=1}^{m} \left( B_\chi^p \right)_{m,l}^2 = S_1 + S_2 = \Theta(P_1 + P_2) \pm \mathcal{O}(Q_1 + Q_2).$$

We have that

$$P_1 = r_{p-1}^2 \sum_{l=1}^{\max\{m-p,0\}} \chi_{l+p-1}^2 = r_{p-1}^2 \sum_{l=p}^{m-1} \chi_t^2 = \Theta\left( \frac{1}{p} \sum_{t=p}^{m-1} \chi_t^2 \right),$$

$$P_2 = \chi_m^2 \sum_{l=q+1}^{m} r_{m-l}^2 = \chi_m^2 \sum_{t=0}^{\min\{m,p\}-1} r_t^2 = \Theta\left( \chi_m^2 \log \min\{m, p\} \right),$$

from using the bound $r_t = \Theta(1/\sqrt{t})$, and

$$Q_1 + Q_2 \leq \sum_{l=1}^{m-1} \left( \sum_{t=0}^{m-l-1} \Delta_{l+t} r_t \right)^2 \leq Q,$$

from increasing the upper limit of the inner sum of $Q_1$ to $m - l - 1$, then setting $m = n$. And so,

$$\sum_{l=1}^{m} \left( B_\chi^p \right)_{m,l}^2 = \Theta\left( \chi_m^2 \log (\min\{m, p\}) + \frac{1}{p} \sum_{t=p}^{m-1} \chi_t^2 \right) \pm \mathcal{O}(Q).$$

Computing the norms:

$$\frac{1}{n} \| B_\chi^p \|_F^2 = \frac{1}{n} \sum_{m=1}^{n} \sum_{l=1}^{m} \left( B_\chi^p \right)_{m,l}^2 = \Theta\left( \frac{1}{n} \sum_{m=1}^{n} \left[ \log \min(\{m, p\}) + \frac{\max\{m - p, 0\}}{p} \right] \right) \pm \mathcal{O}(Q)$$

$$\| B_\chi^p \|_{2 \to \infty}^2 = \max_{1 \leq m \leq n} \sum_{l=1}^{m} \left( B_\chi^p \right)_{m,l}^2 = \max_{1 \leq m \leq n} \Theta\left( \log \min(\{m, p\}) + \frac{\max\{m - p, 0\}}{p} \right) \pm \mathcal{O}(Q)$$

For each of the two norms, the first term has smallest asymptotic growth for $p \sim n$, yielding $\Theta(\log n)$, and so is $\Omega(\log n)$ for all choices of $p$, thus dominating $\pm\mathcal{O}(Q)$. Taking a square-root finishes the proof. $\qquad \square$

**Theorem 3.** *Under the same assumptions on learning rate scheduling $\chi_t$ as in Theorem 1, the following holds.*

$$\mathcal{E}(B^p_\chi, C^p_1) = \mathcal{O}\left(\sqrt{\frac{k}{n}\left(\log p + \frac{p}{b}\right) \sum_{m=1}^{n}\left[\chi_m^2 \log\left(\min\{m,p\}\right) + \frac{1}{p}\sum_{t=p}^{m-1}\chi_t^2\right]}\right) . \tag{23}$$

*Proof.* As shown in the proof of Theorem 1, the condition on $\chi_t$ is sufficient to enforce $Q = o(\log n)$, and so

$$\frac{1}{\sqrt{n}}\|B^p_\chi\|_F = \Theta\left(\sqrt{\frac{1}{n}\sum_{m=1}^{n}\left[\chi_m^2 \log\left(\min\{m,p\}\right) + \frac{1}{p}\sum_{t=p}^{m-1}\chi_t^2\right]}\right)$$

from invoking Lemma 20. For the sensitivity $\mathrm{sens}(C^p_1)$, we use the following bound from (Kalinin et al., 2025, Theorem 2 proof)

$$\mathrm{sens}(C^p_1) = \mathcal{O}\left(\sqrt{k\log p + \frac{kp}{b}}\right) .$$

Inserting the two bounds into $\mathcal{E}(B^p_\chi, C^p_1) = \frac{1}{\sqrt{n}}\|B^p_\chi\|_F \cdot \mathrm{sens}(C^p_1)$ gives the statement. $\qquad\square$

**Corollary 3.** *Let $\chi_t = \beta^{\frac{t-1}{n-1}}$ with $\beta \in (0, 1/e)$. Then, in multi-participation with $b$-min-separation and at most $k = \lceil \frac{n}{b} \rceil$ participations, we have for $p^* \sim b\log b$ the following optimized upper bound:*

$$\mathcal{E}(B^p_\chi, C^p_1) = \mathcal{O}\left(\frac{\sqrt{k}\log n + k}{\sqrt{\log(1/\beta)}}\right) . \tag{24}$$

*Proof.* As $\chi_t$ satisfies the condition of Theorem 3, we have that

$$\mathcal{E}(B^p_\chi, C^p_1) = \mathcal{O}\left(\sqrt{\frac{k}{n}\left(\log p + \frac{p}{b}\right) \sum_{m=1}^{n}\left[\alpha^{2(m-1)} \log\left(\min\{m,p\}\right) + \frac{1}{p}\sum_{t=p}^{m-1}\alpha^{2(t-1)}\right]}\right) ,$$

where $\alpha = \beta^{\frac{1}{n-1}}$. We will evaluate each of the two terms in the outer sum. First off,

$$\sum_{m=1}^{n}\alpha^{2(m-1)} \log\left(\min\{m,p\}\right) \le \log p \sum_{m=1}^{n}\alpha^{2(m-1)} = \Theta\left(\frac{n\log p}{\log(1/\beta)}\right) ,$$

where the last step follows from the proof of Lemma 11. Proceeding with the second term:

$$\frac{1}{p}\sum_{m=1}^{n}\sum_{t=p}^{m-1}\alpha^{2(t-1)} = \frac{1}{p}\sum_{t=p}^{n-1}(n-t)\alpha^{2(t-1)} \le \frac{n-p}{p}\sum_{t=p}^{n-1}\alpha^{2(t-1)} = \mathcal{O}\left(\frac{n^2}{p\log(1/\beta)}\right) ,$$

where the last step again uses the proof of Lemma 11. It follows that

$$\mathcal{E}(B^p_\chi, C^p_1) = \mathcal{O}\left(\sqrt{\frac{k}{\log(1/\beta)}\left(\log p + \frac{p}{b}\right)\left(\log p + \frac{n}{p}\right)}\right) .$$

As this exactly matches the error given in (Kalinin et al., 2025, Theorem 2), up to the $1/\sqrt{\log(1/\beta)}$ factor, the upper bound is minimized for the choice of $p^* \sim b\log b$ achieving error

$$\mathcal{E}(B^p_\chi, C^p_1) = \mathcal{O}\left(\frac{\sqrt{k}\log n + k}{\sqrt{\log(1/\beta)}}\right) ,$$

completing the proof. $\qquad\square$

## H  MULTI-PARTICIPATION: LOWER BOUNDS

**Theorem 4** (Lower bound for multi-participation). *Let $A_\chi = A_1 D_\chi$, where $D_\chi = \mathrm{diag}(\chi_1, \ldots, \chi_n)$ with positive $\chi_t > 0$. Assume any factorization $A_\chi = B \times C$. Then, in multi-participation with b-min-separation and at most $k = \lceil \frac{n}{b} \rceil$ participations, we have*

$$\mathcal{E}(B,C) \geq \max \left\{ \max_{t \leq n} \frac{\sqrt{k}\, t\, \chi_t}{\pi \sqrt{2} n} (\min_{j \leq t} \chi_j) \log(t), \; \sum_{j=0}^{k-1} \chi_{1+jb} \left( 1 - \frac{j}{k-1} \right) \right\}. \tag{25}$$

*Proof.* We start with the first bound, by definition,

$$\mathcal{E}(B,C) = \frac{1}{\sqrt{n}} \|B\|_F \cdot \mathrm{sens}_{k,b}(C). \tag{69}$$

If we restrict to the principal submatrices $B_{:t,:}$ and $C_{:,:t}$, then removing the rows can only decrease the Frobenius norm, and removing the last $n - t$ columns can only decrease the sensitivity, since any participation pattern for the matrix $C_{:,:t}$ would be a valid pattern for the full matrix. Hence

$$\mathcal{E}(B,C) \geq \frac{1}{\sqrt{n}} \|B_{:t,:}\|_F \cdot \mathrm{sens}_{k,b}(C_{:,:t}). \tag{70}$$

Following the proof of Lemma 9 in (Kalinin et al., 2025), we have

$$\mathrm{sens}_{k,b}(C_{:,:t}) \geq \frac{1}{\sqrt{2b}} \|C_{:,:t}\|_F. \tag{71}$$

Therefore,

$$\mathcal{E}(B,C) \geq \frac{1}{\sqrt{2nb}} \|B_{:t,:}\|_F \cdot \|C_{:,:t}\|_F. \tag{72}$$

Applying the Schatten inequality for Frobenius and nuclear norms,

$$\|B_{:t,:}\|_F \cdot \|C_{:,:t}\|_F \geq \|(A_\chi)_{:t,:t}\|_*, \tag{73}$$

which gives

$$\mathcal{E}(B,C) \geq \frac{1}{\sqrt{2nb}} \|(A_\chi)_{:t,:t}\|_*. \tag{74}$$

Finally, by Lemma 19,

$$\|(A_\chi)_{:t,:t}\|_* \geq \frac{1}{\pi} (\min_{j \leq t} \chi_j) t \log t, \tag{75}$$

which implies

$$\mathcal{E}(B,C) \geq \max_{t \leq n} \frac{\sqrt{k}\, t}{\pi \sqrt{2} n} (\min_{j \leq t} \chi_j) \log t. \tag{76}$$

For the second bound, we use the proof of Theorem 1 from Kalinin et al. (2025), which shows that

$$\mathcal{E}(B,C) \geq \frac{1}{\sqrt{n}} \|BC\pi_1\|_2 = \frac{1}{\sqrt{n}} \|A_\chi \pi_1\|_2, \tag{77}$$

where $\pi_1$ is a vector with ones in positions $1 + jb$ for $j \in [0, k-1]$, and zeros elsewhere. We can lower bound the norm explicitly:

$$\frac{1}{\sqrt{n}} \|A_\chi \pi_1\|_2 = \sqrt{\frac{1}{n} \sum_{i=0}^{k-1} \sum_{j=0}^{k-1} \chi_{1+jb} \chi_{1+ib} (n - jb)}$$

$$= \sqrt{\sum_{i=0}^{k-1} \sum_{j=0}^{k-1} \chi_{1+jb} \chi_{1+ib} \left( 1 - \frac{j}{n/b} \right)}$$

$$\geq \sum_{j=0}^{k-1} \chi_{1+jb} \left( 1 - \frac{j}{n/b} \right) \geq \sum_{j=0}^{k-1} \chi_{1+jb} \left( 1 - \frac{j}{k-1} \right),$$

which concludes the proof.

$\square$

**Corollary 4.** *Let* $\chi_k = \beta^{\frac{k-1}{n-1}}$ *with* $\beta \in (0, 1/e)$. *Then Theorem 4 yields*

$$\mathcal{E}(B, C) = \Omega\left(\frac{\sqrt{k}}{\log(1/\beta)}\log\frac{n}{\log(1/\beta)} + \frac{k}{\log(1/\beta)}\right). \tag{26}$$

*Proof.* We substitute $\chi_k = \beta^{\frac{k-1}{n-1}}$ in the general lower bound:

$$\mathcal{E}(B, C) \geq \max\left\{\max_{t\leq n}\frac{\sqrt{k}\,t\,\chi_t\,\log(t)}{\pi\sqrt{2n}}, \sum_{j=0}^{k-1}\chi_{1+jb}\left(1 - \frac{j}{k-1}\right)\right\}. \tag{78}$$

For the first term, we substitute $t = \lceil\frac{n}{\log(1/\beta)}\rceil$, which gives $\chi_t = \Theta(1)$, resulting in

$$\mathcal{E}(B, C) = \Omega\left(\frac{\sqrt{k}}{\log(1/\beta)}\log\frac{n}{\log(1/\beta)}\right). \tag{79}$$

The second term, we compute explicitly:

$$\sum_{j=0}^{k-1}\chi_{1+jb}\left(1 - \frac{j}{k-1}\right) = \alpha\frac{\alpha^{bn} + (1-\alpha^b)n - 1}{(1-\alpha^b)^2(n-1)}, \tag{80}$$

where $\alpha = \beta^{1/(n-1)}$. Asymptotically this is equal to $\frac{1}{1-\alpha^b}$, giving the lower bound

$$\mathcal{E}(B, C) = \Omega\left(\frac{n}{b\log(1/\beta)}\right) = \Omega\left(\frac{k}{\log(1/\beta)}\right). \tag{81}$$

Combining those lower bounds as an average, we conclude the proof. $\square$

