# OpenReview forum: "Learning Rate Scheduling with Matrix Factorization for Private Training"
_ICLR.cc/2026/Conference — ICLR 2026 Conference Withdrawn Submission_

### Official Review · Reviewer_M91X · 2025-10-30

**Soundness:** 3
**Presentation:** 3
**Contribution:** 3
**Rating:** 6
**Confidence:** 2

**Summary:**

The paper studies differentially private training with non-constant learning-rate schedules (LRS) using matrix‑factorized correlated noise. It (i) introduces a schedule‑aware factorization $B_\chi A_1^{1/2}$ and proves single‑participation bounds for $\mathrm{MaxSE}$ and $\mathrm{MeanSE}$ under mild schedule smoothness (Theorem 1), with schedule‑specific rates for exponential, polynomial, linear, and cosine decays (Appendix E). (ii) It gives single‑participation lower bounds (Theorem 2) matching the upper‑bound structure up to constants. (iii) For multi‑participation with $b$-separation, it presents a general upper bound (Eq. (23)) and an exponential‑schedule corollary optimized at $p^* \sim b\log b$ (Corollary 3), together with a general lower bound (Theorem 4) and an exponential corollary (Corollary 4). (iv) Experiments (Section 5) on CIFAR‑10 with a 3‑block CNN under $(9,10^{-5})$-DP illustrate utility gains from LRS and additional gains from LRS‑aware factorizations, with comparisons across memory‑efficient mechanisms (“w/ LRS” vs “w/o LRS”).

**Strengths:**

- **Clear theoretical framing (Section 2; Theorem 1):** The schedule‑aware factorization $B_\chi A_1^{1/2}$ is analyzed under two verifiable smoothness conditions, either $\Delta_t\le c/(t(1+\log t))$ or $\sum_t \Delta_t^2 = o(\log n/n)$, yielding explicit $\mathrm{MaxSE}$ and $\mathrm{MeanSE}$ rates as functions of the schedule $\{\chi_m\}$ and $\log n$.
- **Tightness across regimes (Appendix F/H):** Single‑participation lower bounds (Theorem 2) and multi‑participation lower bounds (Theorem 4) mirror the structure of the corresponding upper bounds; exponential‑schedule corollaries make the dependences on $k, b$, and $\log(1/\beta)$ explicit.
- **Breadth of schedules (Appendix E):** Appendix E derives schedule‑specific bounds and supporting lemmas (e.g., exponential, cosine), consolidating applicability beyond exponential decay.
- **Actionable empirical narrative (Section 5):** Section 5 compares “w/ LRS” vs “w/o LRS” across several memory‑efficient factorizations and shows consistent improvements with LRS, aligning practice with the theoretical claims.

**Weaknesses:**

- **Section 5: the privacy accounting needs to be harmonized and the $\varepsilon$ per setting must be reported.** Standardizing the privacy accountant across all compared methods and explicitly reporting the resulting $(\varepsilon,\delta)$ to attribute accuracy deltas to LRS/mechanism rather than accounting differences will help strengthen the paper. This could be done by adding a compact table in Section 5.
- **Section 5: compute/memory overheads of LRS‑aware mechanisms.** Adding wall‑clock time, peak memory/VRAM, and per‑step FLOP estimates for Banded/BandInv/BISR (“w/ LRS” and “w/o LRS”) will help quantify the cost–utility trade‑off, since Section 5 already contrasts these classes.
- **Appendix H (Theorem 4 → Corollary 4).** To improve readability, the paper could provide an explicit lemma justifying the choice of the optimizing index (the selected $t$) and the transition to the stated $\Omega\big(\sqrt{k}\,\tfrac{\log n}{\log(1/\beta)} + \tfrac{k}{\log(1/\beta)}\big)$ form of the lower‑bound derivation.
- **Scope (Section 5):** The current empirical results are only on CIFAR‑10 with a small CNN. Adding at least one larger architecture (e.g., ResNet‑18/34) or a language task will demonstrate the robustness across modalities.

**Questions:**

1) **Section 5:** Does the relative ordering among “w/ LRS” vs “w/o LRS” mechanisms persist under a single, unified privacy accountant and identical clipping/Poissonization assumptions? Please include a table with per‑setting $\varepsilon$.
2) **Section 5 / Figures:** Since MaxSE curves include a labeled "lower bound” overlay, can a practical spectral/heuristic estimator for attainable MaxSE be added to contextualize the gap between lower bounds and realized mechanisms?
3) **Section 5:** Adding an ablation connecting RMSE to test accuracy across “w/ LRS” vs “w/o LRS” to delineate when optimizing for the LRS workload yields accuracy gains and when it plateaus will be beneficial.
4) **Points mentioned in the weaknesses** Addressing some of the points mentioned in the weaknesses (specifically the ones around Section 5) might strengthen the paper.

---

### Official Review · Reviewer_yF2J · 2025-11-01

**Soundness:** 3
**Presentation:** 3
**Contribution:** 3
**Rating:** 6
**Confidence:** 2

**Summary:**

This paper studies the impact of different learning rate schedulers on the convergence of DP-SGD for matrix factorization, deriving lower and upper bounds for MaxSE and MeanSE in single- and multi-epoch settings, and showing that exponentially decaying learning rates are optimal for MaxSE in single-epoch scenarios. Extensive experiments demonstrate improved convergence under the proposed schedulers.

**Strengths:**

- This paper investigates the use of various popular learning rate schedulers, beyond fixed learning rates, to enhance the convergence of DP-SGD in matrix factorization tasks.

- The authors derive lower and upper bounds for MaxSE and MeanSE in matrix factorization under a class of commonly used learning rate schedulers, for both single-epoch and multi-epoch settings. Notably, exponentially decaying learning rates are shown to be optimal for MaxSE in single-epoch scenarios and also improve the upper bound for MeanSE.

- Extensive experiments compare different learning rate strategies, demonstrating that the proposed schedulers effectively improve the convergence performance of DP-SGD for private optimization problems.

**Weaknesses:**

The experiments were conducted using only a single choice of $(\epsilon, \delta)$-DP parameters and a single benchmark dataset. It would be more comprehensive to evaluate the algorithms across multiple DP parameter settings and datasets. Additionally, since $\epsilon = 9$ corresponds to a relatively weak privacy guarantee, it is recommended to test stricter privacy regimes, such as $\epsilon \in [1, 3]$.

**Questions:**

- The symbol $B$ is used inconsistently, representing both the batch size in Algorithm 1 and a matrix in Equation (2).

- In the first paragraph of Section 5, it would be helpful to provide more details about the proposed algorithms as well as the baseline algorithms used for benchmarking or empirical evaluation.

---

### Official Review · Reviewer_CodA · 2025-11-01

**Soundness:** 3
**Presentation:** 2
**Contribution:** 2
**Rating:** 4
**Confidence:** 3

**Summary:**

This paper studies matrix factorization mechanisms for differentially private learning with variable learning-rate schedules.
The authors introduce a Toeplitz workload that incorporates learning-rate decay and show asymptotic optimality for MaxSE under exponential schedules.
The authors perform experiments on CIFAR-10 to test the practical impact of their approach.

**Strengths:**

* **Important research gap.**
You address an important research gap in private learning.
The idea to investigate matrix factorisation for private learning with variable learning rate is very relevant as, as you mention, learning rate schedulers are widely used.
Your contribution is clearly situated among related work.

* **Interesting formulation via Toeplitz matrices with provable optimality.**
The result is cleanly framed and well organized.
The use of a Toeplitz workload matrix to take learning rates into account is interesting and novel.
For the specific setting of exponential decay and for the MaxSE error, the bounds obtained using the learning-rate-aware Toeplitz workload is shown to be asymptotically optimal.
Even though I did not check proofs in detail, I did not find any glaring mistake.

* **Experiments with comprehensive baselines.**
The experiments you perform compare your factorization approach together with recent matrix-mechanism variants, as well as with DP-SGD.
I think the methodology followed in the experiments is, on a high level, fair and that the comparisons are meaningful.
The experiments are also well documented, helping reproducibility.

**Weaknesses:**

* **Experimental results and weak privacy regime.**
Even though, as mentioned, you perform cleanly presented experiments with a good number of baselines, I think that your experimental setting should be improved.
While I understand that this is mostly a theoretical contribution, the privacy setting chosen $\epsilon=9$ is at the margin of what is considered to be private in practice.
If I understand correctly, the effects of appropriate learning rate scheduling, and therefore of a factorization that takes that into account, should be especially visible in low-noise regimes, such as the ones you experiment with.
Given that the empirical impact of your approach appears to be very limited even with weak privacy guarantees, I wonder if your proposed factorization is impactful at all at stronger privacy regimes (e.g., $\epsilon \approx 1$).
In particular, the learning-rate-aware factorization you propose sometimes performs _worse_ than the simpler baselines.

* **Visualizations.**
Overall, the visualizations should be significantly improved as they are difficult to parse and interpret.
In Figure 2, the red line is essentially invisible in the plot, which makes it impossible to see how the approach in question performs.
In Figure 3b, it is extremely hard to distinguish the dashed lines and, in general, the various symbols.
Moreover, the legend obstructs a part of the plot.
The use of colors is also not consistent between plots and can cause confusion.

* **No clear link between MeanSE/MaxSE and practical improvements.**
The results you provide do not discuss the link between the better theoretical bounds on MeanSE/MaxSE and measurable quantities such as accuracy.
Your plots, with the caveats above, show that your approach does indeed lead to better/smaller MeanSE (Figure 2) when compared to alternative ones.
However, as previously mentioned, this does not reflect in practical improvements.


* **Unclear impact of constants.**
While you show that your approach obtains better/optimal asymptotic results, the impact of constants remains unclear to me.
If I am not misunderstanding, the best result you obtain with Toeplitz factorization (line (e) in Table 2) amounts to roughly a $\sqrt(\log(1/\beta))$ term at the denominator: how impactful can this be expected to be in practice?


### Minor remarks
* $Z$ is not defined in Equation 2.

**Questions:**

* How can you argue for a meaningful practical benefit of your approach?
* Is your improvement meaningful in stronger privacy regimes?

---

### Official Review · Reviewer_v5zw · 2025-11-02

**Soundness:** 3
**Presentation:** 2
**Contribution:** 2
**Rating:** 2
**Confidence:** 4

**Summary:**

In the setting of differentially private learning with correlated noise using matrix factorizations, the paper proposes to factorize a workload matrix dependent on the learning rate. The paper gives bounds on the error properties of some such factorizations, demonstrating an improvement over baselines. The paper also considers private deep learning experiments with learning rate-aware factorizations.

**Strengths:**

The main paper is well-written, it is easy to read and clear to understand (although the proofs are terse and hard to parse, more on that later). I could not spot a single typo in the paper. The importance of learning rate scheduling in private training is well justified. The baselines and proposed approaches are generally sensible.

**Weaknesses:**

* **Theoretical novelty**: Theoretical novelty appears to be limited. The proof techniques and methods are, to a large extent, based on the prior literature. Example: the use of generating functions was common from at least as far back as [Fichtenberger et al.](https://arxiv.org/pdf/2202.11205).
* **Empirical impact**: The empirical impact appears to be limited as well. In Figure 3b, there does not appear be any noticeable difference between matrix factorization based on the agnostic workload $A_1$ and the proposed learning rate-dependent workloads. The improvements in fig 3c also do not seem to be insignificant (aside: this figure is hard to interpret, it may be easier to read as a table). Similarly in Figure 2, BISR without the proposed LRS appears to be better than BISR with LRS. This makes me question the fundamental importance of the proposed LRS. In fact, these results corroborate the recommendation in Section 4.4.1 of [Pillutla et al](https://arxiv.org/pdf/2506.08201), which is to just use the independent workload $A_1$ regardless of the base optimizer or learning rate. (This is also acknowledged in line 431a)
* **Impractical assumptions**: Since there is a decay every step, $\beta \le 1/e$ seems impractical. Within 10 steps, the learning rate reduces to $1/e^{10} \approx 4.5 \times 10^{-4}$ its initial value, which is unreasonably small. It seems to me that this assumption is in place due to the $\log\log(1/\beta)$ factor in the bound, making it a theoretically necessary but practically unreasonable.
* **Insignificant gap**: How significant is the gap between Corollary 1 & Lemma 2/Corollary 2? The learning rate-aware factorization improves $\log n$ to $\log n - \log\log(1/\beta)$. As argued above, this $\log\log(1/\beta)$ factor cannot be large in any practical regime, making the learning rate-independent naive baseline practically just as good.
* **Constants**: This work includes the practically irrelevant $\log\log(1/\beta)$ factor but ignores potentially significant leading constants. (E.g. [Fichtenberger et al.](https://arxiv.org/pdf/2202.11205) show that constants are crucial in matrix factorization)
* **Terse Proofs**: I found the proofs very hard to follow (especially giving the short reviewing cycles) and I couldn't verify some of the claims.
     - I do not understand how some steps follow. Example: eq. 52, $\sum_u r_u^2$ just before eq 54, eq 55, the last few lines of the proof of lemma 11 (which also appear very handwavy) .
     - some functions/quantities are used without introduction. A quick recap can help. Examples: q-gamma function, q-Pochhammer symbol, q-binominal theorem, some generating function properties, Young's convolution identity, etc.
     - Careful and detailed proof-writing without any skipped steps will greatly help.

Minor remarks:
* Eq 4: make MaxSE appears to be square root of maximum expected squared error in any step. That is, the max is outside the expectation, rather than inside as implied by Eq 4. Please check
* Section 4.1 does not compare to any baselines. Running the methods of Section 4 + composition seems like a baseline worth considering.

Some missing citations:
* https://arxiv.org/pdf/2506.08201
* https://arxiv.org/pdf/2502.06597
* https://arxiv.org/pdf/2310.06771
* https://arxiv.org/pdf/2302.01463

**Questions:**

* Fig 3: why isn't there a baseline of BLT with LRS? Seems like it would be straightforward to incorporate it given all the other experiments already given.
* How does the accounting with amplification work for matrix factorization mechanisms? What sample size is used for the Monte Carlo accounting? If public code is used for this, it would be good to point out which one (or conversely, if the authors use their own implementation, they should mention that as well)

---

### Note · Authors · 2025-11-12

**Comment:**

We thank the reviewers for their careful reading and evaluation of our work. In light of the reviewers’ current scores, we find acceptance improbable, and so we opt for withdrawing our submission. We comment on and rebut some of the main concerns below.

**Paper Revision.**
Before, however, we would like to mention that in an upcoming revised version of the paper, we add a new experiment: fine-tuning BERT-base for sentiment analysis on the IMDB dataset with $\varepsilon= 4$, cover a different model, a more challenging setting, and a lower-privacy regime. We will also address comments from the reviewers.

**Meaning of $\beta$ and its range.**
$\beta$ is defined as the learning rate at the last gradient update for learning rate decay; it does not reflect a decay per step. This is stated at the start of Section 3. Enforcing $\beta < 1/e$ is therefore a mild and realistic restriction.

**Leading constants in error bounds.**
We agree that our error bounds (often tightly) express the asymptotic dependence on $n$ and $\beta$. Leading constants in our bounds for different factorizations, errors, and learning rates, can be inferred from our proofs, but we make no claim that these constants are tight. Our concern is instead identifying the sensitivity of different factorization wrt $\beta$. In terms of finer comparison, Figure 1 empirically demonstrates the gain in the error from designing factorizations that take learning rate into account, in the case of exponential learning rate decay.

In the work of Fichtenberger, the workload matrix is the prefix sum matrix, which does not have any parameters other than its size. Therefore, it was natural to explore the constant factors and, due to the simplicity of this workload matrix, even the next term. However, once the workload becomes more complex, for example in cases involving exponential decay, or momentum and weight decay (see Kalinin and Lampert, 2024), none of the constants are known, and even the parameter dependencies remain unclear. We argue that the workload with different learning rates is at least one level more complex than any workload studied before. Therefore, we begin with the most natural problem, identifying the asymptotic dependence on its parameters. Deriving tight leading constants for the error bounds of learning-rate-aware factorizations is left as (non-trivial) interesting future work.

**Empirical impact.**
We acknowledge that, based on the presented plots, the performance difference between the proposed factorization (BISRwLR) and the standard BISR is not visually large, yet it is statistically significant. The main contribution of our work lies in providing a detailed error analysis for workload matrices with varying learning rates. We demonstrate that BISRwLR has provably lower MeanSE than BISR. However, as also noted by Pillutla et al., this theoretical improvement does not always translate directly into higher accuracy.

**Missing references.**
We thank for pointing out that we haven’t cited the survey Pilluta et al. For the rest of the papers, we want to ensure that we are aware of them. There are many other papers on matrix factorization, however we do not find them to be relevant enough to be cited.

**BLT as baseline**
To the best of our knowledge, BLT has only been described as an optimization procedure for the prefix-sum workload, and, in a very recent paper [1], for workloads with momentum. We find it nontrivial to efficiently implement BLT for workloads with different learning rates. Moreover, we rely on the jax-privacy scalable implementation, which does not support other workloads. We consider this an interesting direction for future research.

[1] Huang et al. “Memory-Efficient Correlated Noise for Locally Differentially Private Momentum in Distributed Learning” 2025

**Withdrawal Confirmation:**

I have read and agree with the venue's withdrawal policy on behalf of myself and my co-authors.